# Non-metallic T$_2$-MRI agents based on conjugated polymers

Qinrui Lin[1,2], Yuhong Yang[3✉] & Zhengzhong Shao [1,2,4,5✉]

Developing non-metallic contrast agents of clinically applied magnetic resonance imaging (MRI) is an alternative strategy to reduce the toxicity of heavy metal elements in current MRI agents. These non-metallic MRI agents usually generate contrasts by unpaired electrons, which are prone to be deactivated by in vivo radical scavenging pathways. Since the unpaired electrons in conjugated polymers exhibit satisfying stability for in vivo imaging, developing conjugated polymers based MRI agents may solve the in vivo stability problem of current non-metallic agents. However, MRI-active properties have not been reported in existing conjugated polymers yet. Herein we report on MRI-active conjugated polymer nanoparticles based on polypyrrole (PPy), which can be used for in vivo imaging. Our method not only introduce a kind of non-metallic MRI agents but extends the applications of conjugated polymers from optical imagings to MRI.

[1] State Key Laboratory of Molecular Engineering of Polymers, Fudan University, Shanghai, China. [2] Laboratory of Advanced Materials, Fudan University, Shanghai, China. [3] Research Center for Analysis and Measurement, Fudan University, Shanghai, China. [4] Department of Macromolecular Science, Fudan University, Shanghai, China. [5] Jiangsu Collaborative Innovation Center of Biomedical Functional Materials, Nanjing Normal University, Nanjing, China. ✉email: yuhongyang@fudan.edu.cn; zzshao@fudan.edu.cn

Magnetic resonance imaging (MRI) has been extensively applied in diagnosis because it can provide detailed anatomical pictures without concerning ionization irradiation and tissue-penetration depth[1–3]. The contrasts of clinically applied MRI are originated from relaxation times of water protons[3]. To create contrast between naive water and target tissues, the relaxation times of target tissues are changed by introducing contrast agents.

Current contrast agents for clinically applied MRI techniques are divided into $T_1$ and $T_2$ agents. $T_1$ agents contain paramagnetic metal ions that can interact with water protons to shorten their longitude relaxation times[4]. Meanwhile, $T_2$ agents are superparamagnetic nanoparticles to create a magnetic inhomogeneity to shorten the transverse relaxation time of water protons[5]. Both $T_1$ and $T_2$ MRI agents usually contain heavy-metal elements, which can induce failures in the liver and kidney[6–8]. Even for the FDA-approved MRI agents, superparamagnetic iron oxide nanoparticles have been reported to cause fatal allergic reactions[9–11]. For alleviating these biosafety concerns, an alternative strategy is developing organic agents that generate MRI contrast without metal ions[9].

The majority organic contrast agents for MRI are applicable in some newly developed MRI techniques, including [19]F-MRI, CEST, and hyperpolarization[12–14]. Although these techniques exhibited improved imaging qualities, their requirements for developing biosafe and clinically applicable instruments constrained these organic agents' translation into clinics[3,9,15]. Until now, the organic agents that can be used in clinically used MRI techniques are phenols and nitroxide radical molecules[16,17]. Only nitroxide radical molecules exhibited potential for in vivo imaging. The MRI properties of nitroxide radical molecules come from their unpaired radical electrons, which are stabilized by steric hindrance. However, nitroxide radicals suffer from poor in vivo stability. This is because the paramagnetic radicals can be rapidly deactivated by in vivo radical scavenging pathways[9]. Current approaches to increase the in vivo stability of nitroxide radicals are elaborating the chemical structures by tedious synthetic processes. Thus, very few nitroxide radical-based MRI agents exhibit "MRI-active" state for hours. Therefore, further developing organic MRI agents still requires finding a new strategy to stabilize paramagnetic unpaired electrons.

We noticed that oxidizing conjugated polymers can also generate paramagnetic unpaired electrons[18,19]. These unpaired electrons are stabilized by delocalization along the conjugated backbone. The optical properties of these unpaired electrons can be used for in vivo imaging and therapy. Hence, conjugated polymers may be a kind of organic MRI agents with increased in vivo stability of unpaired electrons. Although previous studies have discovered the paramagnetism in existing conjugated polymers, organic MRI agents based on conjugated polymers have not been reported yet. How to utilize the paramagnetism of conjugated polymers to develop organic MRI agents remains to be explored.

In this work, we propose a "partially complexed" method for turning paramagnetic conjugated polymers into $T_2$ MRI agents, such as a commonly used conjugated polymer (polypyrrole, PPy). We investigate the correlations between unpaired electrons and the $T_2$ MRI contrast of PPy nanoparticles, and their biosafety and feasibility for in vivo imaging as well. Moreover, some preliminary general applicability of this method is also conducted using other similar conjugated polymers, like poly (3,4-ethyl-dioxythiophene) (PEDOT) and polyaniline (PANI).

## Results and discussion

In this research, PPy nanoparticles in about 15-nm diameter (Fig. 1a) are successfully synthesized through a modified chemical oxidation method, called "partially complexed". In this method, the oxidant ferric ions were partially complexed with catechol derivatives (dopamine, DA; catechol, CA; 2, 3-dihydroxybenzoic acid, DHBA; and 3,4-dihydroxybenzonitrile, DHBN). As-prepared catechol–PPy nanoparticles exhibit exceptional $T_2$ MRI contrasts, which may be generated by three factors: residual iron elements brought by ferric chloride, diamagnetic catechol derivatives, and paramagnetic oxidized conjugated polymers. Therefore, the correlations between the above three factors and $T_2$ MRI contrasts of PPy were first evaluated in the following paragraphs.

The impact of residual iron elements brought on $T_2$ MRI contrast was first excluded by analyzing the iron content. Since ferric chloride was used as the oxidant to prepare both pristine and catechol–PPy nanoparticles, residual metal elements may form negligible inorganic compounds that contributes to the $T_2$ MRI contrast in PPy nanoparticles. Here, the level of residual metal elements was evaluated by thermal decomposition of PPy nanoparticles in air. This thermogravimetric analysis reveals that decomposition product (maybe the oxides formed by residual iron elements) remains at a level of only about 1–2% for both pristine and catechol–PPy nanoparticles (Supplementary Fig. 1a). These results indicate the very low levels of iron elements in all PPy nanoparticles. Besides TGA analysis, XPS spectra of PPy nanoparticles also reveals very low levels of iron elements. The characteristic peak of Fe $2p$ cannot be identified in the survey XPS and narrow-scan spectra for all five kinds of PPy nanoparticles (Supplementary Fig. 1b, c and Supplementary Table 1). Hence, such a low content of metal elements is likely to form only negligible levels of $T_2$ MRI-active inorganic compounds. Hence, the characteristic sharp peaks of inorganic compounds were not observed in the XRD patterns of catechol–PPy nanoparticles (Supplementary Fig. 1d). Although the contents of residual iron elements were similar for pristine PPy and catechol–PPy nanoparticles, only catechol–PPy nanoparticles exhibited $T_2$ MRI contrast. Therefore, residual metal elements might contribute to the $T_2$ MRI properties of catechol–PPy nanoparticles, but they are not the determining factor for generating $T_2$ MRI contrast in catechol–PPy nanoparticles.

Next, the impact of incorporated catechol derivatives should be excluded because they have been reported to generate $T_2$ MRI contrast using a high-field (9.4 T) MRI system[17]. Therefore, we analyzed the content of catechol derivatives in our catechol–PPy nanoparticles, as well as the $T_2$ relaxation times of catechol derivatives using a low-field (0.5 T) MRI system. First, the amounts of catechol derivatives in as-synthesized catechol–PPy nanoparticles were extremely low according to their FTIR spectra (Supplementary Fig. 2a). This is because the FTIR spectra of pristine PPy and catechol–PPy nanoparticles are identical and the characteristic peak of the C=C stretching bond of the benzoic group around $1625 \, cm^{-1}$ was not observed in any of the catechol–PPy nanoparticles. Also, catechol derivatives produce little shortening effect of the $T_2$ relaxation times under a low-field (0.5 T) MRI system used in our research (Supplementary Fig. 2b). After excluding the possibility of iron elements and catechol derivatives for generating $T_2$ MRI contrast, we speculated that the $T_2$ MRI properties can only be activated by the paramagnetism of PPy itself. The paramagnetic species in PPy and other conjugated polymers are polymer radical cations called polarons. Polarons are formed when oxidative agents remove an electron from the valence band of conjugated polymers. In the "partially complexed" method, ferric chloride not only triggers the polymerization of PPy but oxidizes PPy during polymerization.

To prove our speculation, we started to characterize the content and status of polarons in pristine and catechol–PPy nanoparticles. This is because these two factors determine the contrast

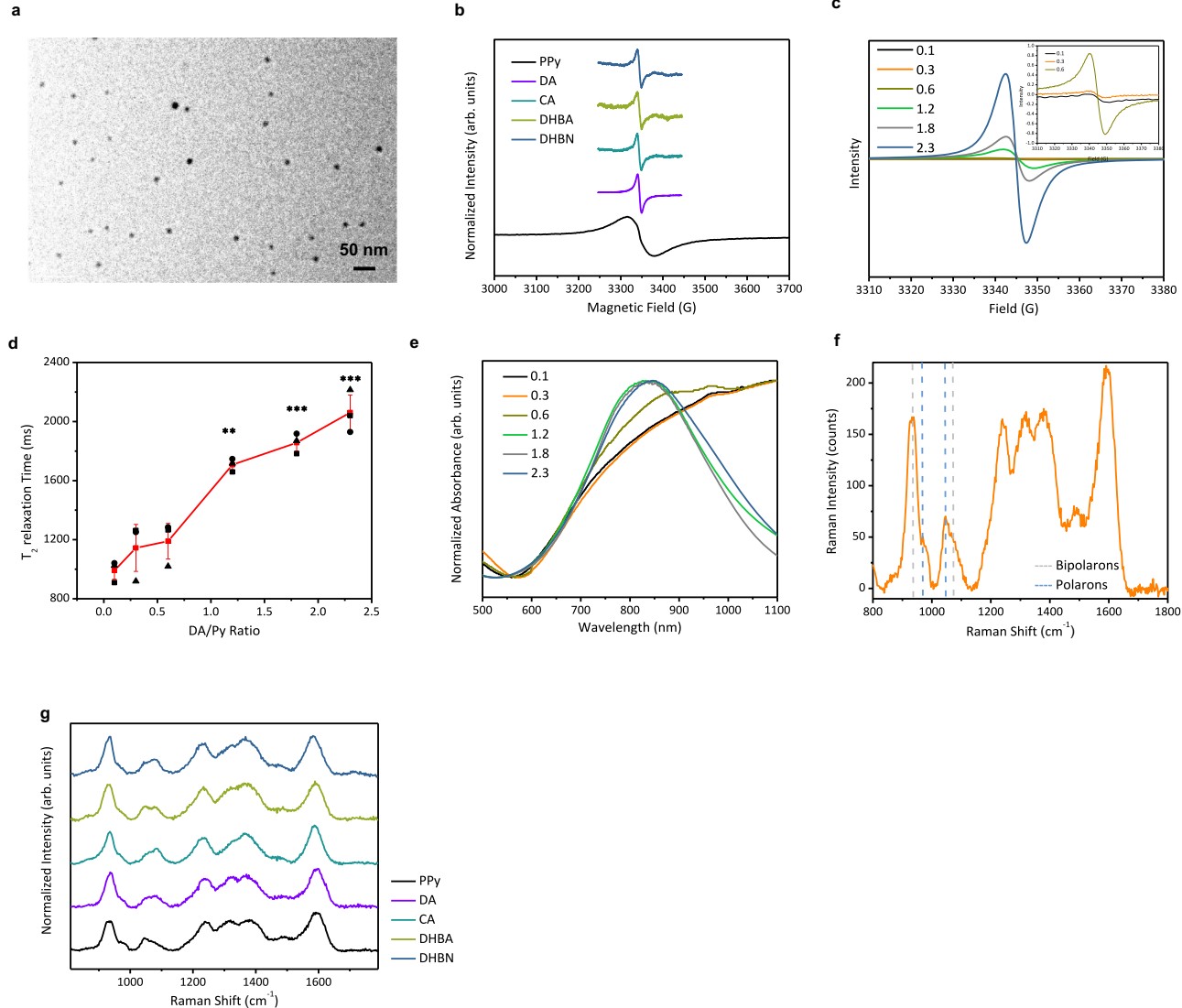

**Fig. 1 Investigating the T$_2$ MRI properties in PPy nanoparticles. a** A representative transmission electron microscopy (TEM) image of catechol–PPy nanoparticles. Experiments were performed three times with similar results. **b** ESR spectra of pristine and catechol–PPy nanoparticles. All spectra were measured in dry powders of PPy nanoparticles. **c** ESR spectra of DA-PPy nanoparticles prepared with different amount of dopamine. All spectra were measured in dry powders of PPy nanoparticles. **d** T$_2$ relaxation times of 1 mg/mL DA–PPy nanoparticles in aqueous solutions. Error bars in d represent mean±s.d., with n = 3 independent experiments. Statistical significance for comparison of 0.1 DA–PPy and other DA–PPy nanoparticles was determined by two-tailed Student's $t$-test (*$p < 0.05$, **$p < 0.01$, and ***$p < 0.001$). $p_{0.3} = 0.195$, $p_{0.6} = 0.633$, $p_{1.2} = 5.66E-05$, $p_{1.8} = 5.13E-05$, $p_{2.3} = 1.48E-03$.
**e** Normalized absorption spectra of DA–PPy nanoparticles prepared with different amount of dopamine in aqueous solutions. **f** A representative Raman spectrum of PPy nanoparticles (acquired with 785-nm laser). The fluorescence background was removed from the spectrum with a polynomial baseline on fourth-order. **g** Raman spectra of pristine and catechol–PPy nanoparticles. Catechol–PPy nanoparticles were prepared with different amount of catechol derivatives. The presented Raman spectra of each kind of catechol–PPy nanoparticles were acquired using the nanoparticles with the highest $R$-value.

of MRI agents[20]. We first tried to correlate the T$_2$ MRI contrast with polaron content in as-synthesized PPy nanoparticles. The polaron contents were estimated by electron-spin spectroscopy (ESR)[21,22] (Table 1). The T$_2$ MRI contrasts were evaluated using r$_2$ relaxivities of PPy nanoparticles, which were calculated as the relaxation rates of T$_2$ relaxation times vs mass concentrations of contrast agents (Table 1). Contrast agents with greater r$_2$ relaxivities can provide enhanced MRI contrast[23]. However, these two parameters do not show any obvious correlation. The polaron content in pristine PPy nanoparticles is one order higher than catechol–PPy nanoparticles but only exhibited negligible r$_2$ relaxivity. Hence, activating the T$_2$ MRI properties of catechol–PPy nanoparticles is also related to the variation of polaron status, which was subsequently characterized using the

linewidth of ESR signals[24]. Significant variation of linewidth in pristine and catechol–PPy nanoparticles reveals that the polaron status in these two kinds of nanoparticles is different (Fig. 1b). According to previous researches[25,26], polarons with broad ESR linewidth are localized on longer polymer chains. Such polarons are in strong spin–spin interactions with each other. While polarons with narrow linewidth exhibit weak spin–spin interactions with other polarons. Such weak spin–spin interactions may be caused by catechol derivatives because they can trap radicals like polarons. Under weak spin–spin interactions, polarons can interact with water protons, which is the basis of contrast-enhanced MRI.

To further get insight into how polaron status varied, we used DA–PPy nanoparticles to investigate the structural parameters

**Table. 1 Different parameters of pristine PPy and PPy nanoparticles prepared with different catechol derivatives.**

| | [Bipolarons]/[polarons] ratio (R)[a] | Polaron content (/mg)[b] | ΔHpp (G) | T2 relaxation time of 1.5 mg/mL CPNs (ms) | r2 relaxivity of CPNs (mL/mg·s) | r2 relaxivity of polarons (L/mmol·s)[c] | Substitutional group |
|---|---|---|---|---|---|---|---|
| DA-PPy (dopamine) | 2.32 | $4.51 \times 10^{14}$ | 9.57 | 847 | 0.467 | 623 | -NH₂ (electron donating) |
| CA-PPy (catechol) | 2.43 | $2.53 \times 10^{14}$ | 9.18 | 962 | 0.438 | 1042 | None |
| DHBA-PPy (2,3-dihydroxybenzoic acid) | 2.95 | $1.651 \times 10^{13}$ | 9.75 | 971 | 0.415 | 15132 | -COOH (electron withdrawing) |
| DHBN-PPy (3,4-dihydroxybenzonitrile) | 2.52 | $8.75 \times 10^{13}$ | 10.05 | 1754 | 0.131 | 901 | -CN (strong electron withdrawing) |
| Pristine PPy | 1.57 | $7.12 \times 10^{15}$ | 45 | 2325 | 0.0358 | 3.02 | / |

[a]The R-values exhibited similar variation trend for different PPy nanoparticles.
[b]The spin counting tool of the Xepr software was used to determine the spin density (polaron amount) of PPy samples after a factory calibration of the cavity-response function. The uncertainty of the spin concentration was estimated to be within ~10%.
[c]$r_2$ relaxivity of polarons were estimated from the spin densities, which were characterized in dry powders, while $T_2$ relaxation times of PPy nanoparticles were measured in aqueous solution. Therefore, $r_2$ relaxivity was only used to investigate the polaron status.

that may be changed in catechol–PPy nanoparticles prepared with our "partially complexed" method. Different DA–PPy nanoparticles were prepared with different fractions of ferric ions complexed with DA (DA/Fe³⁺). With DA/Fe³⁺ increases, the polaron content also increases because the ESR signal intensity of these nanoparticles elevated (Fig. 1c). However, only 0.1/0.3/ 0.6DA–PPy nanoparticles exhibit significant shortening effect on the $T_2$ relaxation times (Fig. 1d). Such inconsistence indicate that the variation of polaron status is further related to other structural parameters. Besides the variation of the ESR signal intensity, the absorption spectra of different DA–PPy nanoparticles also change obviously in the near-infrared region (Fig. 1e). The relative absorbance in the second near-infrared (NIR-II) region (>1000 nm) versus that in the NIR-I region (700–1000 nm) of 0.1/0.3/0.6DA–PPy nanoparticles is higher than that of 1.2/1.8/ 2.3DA–PPy nanoparticles. For PPy, the NIR-I absorbance is assigned to polarons and NIR-II absorbance is assigned to bipolarons. Bipolarons are polymer dications formed by two coupled polarons, when polaron concentration increases.[27] Since bipolarons exhibit two positive charges, the electrostatic repulsion interactions between polarons (cation) and bipolarons (dication) might change the polaron status.

To prove the above speculations, we prepared PPy nanoparticles using different catechol derivatives. Also, two parameters are introduced to investigate the semiquantitative correlation between bipolaron and polaron status. The polaron status is reflected by its $r_2$ relaxivities, which were calculated as the relaxation rates of $T_2$ relaxation times vs molar concentrations of polarons (Table 1). Meanwhile, the relative content of bipolarons vs polarons is estimated using Raman spectroscopy (Fig. 1f). This relative content (denoted as R-value) can be calculated as the ratio between the intensities of the bands relative to bipolarons (934 and 1086 cm⁻¹) and polarons (968 and 1055 cm⁻¹)[28]. The R-value is used to represent the variation trend of the relative content of bipolarons vs polarons, rather than the actual ratio in PPy nanoparticles. As expected, these relative contents appear in the same variation trend with the $r_2$ relaxivities of polarons. The R-value of pristine PPy nanoparticles is only 1.57, but those of catechols–PPy nanoparticles are above 2 (Fig. 1g and Table 1). These observations confirm that variation in the relative content of bipolarons changed the status of the paramagnetic polarons. The increased relative bipolaron content enhances the electrostatic repulsion around polarons. Thus, the interactions between polarons are further reduced and interactions between polarons and water protons are further elevated. According to the above studies, we conclude that the $T_2$ MRI contrasts of catechol–PPy nanoparticles are associated with two steps (Fig. 2): (1) polarons should be released from the strong "spin–spin" interactions with other polarons. In this research, catechol derivatives can fulfill this requirement by their radical trapping ability. (2) Bipolarons are required to be formed around polarons to further enhance the interactions between polarons and water protons. The mechanism of the "partially complexed" method regulating bipolaron content remains to be investigated.

Besides $T_2$ MRI, the increased amount of bipolaron can also be used to photoacoustic (PA) imaging in the second near-infrared window (1000–1700 nm). This is because the absorbance of PPy's bipolaron band lies in this region[29]. The maximum photoacoustic signal of different catechol–PPy nanoparticles appeared around 1280 nm, which is in agreement with the absorbance band of bipolarons (Supplementary Fig. 3a, b). This maximum photoacoustic signal also exhibited in DHBA–PPy nanoparticles (Supplementary Fig. 3a). The similar variation was also observed between the photoacoustic signal at 1280 nm and the R-values (Table 1). The above results confirmed that PA contrast in the NIR-II region of catechol–PPy nanoparticles was generated by

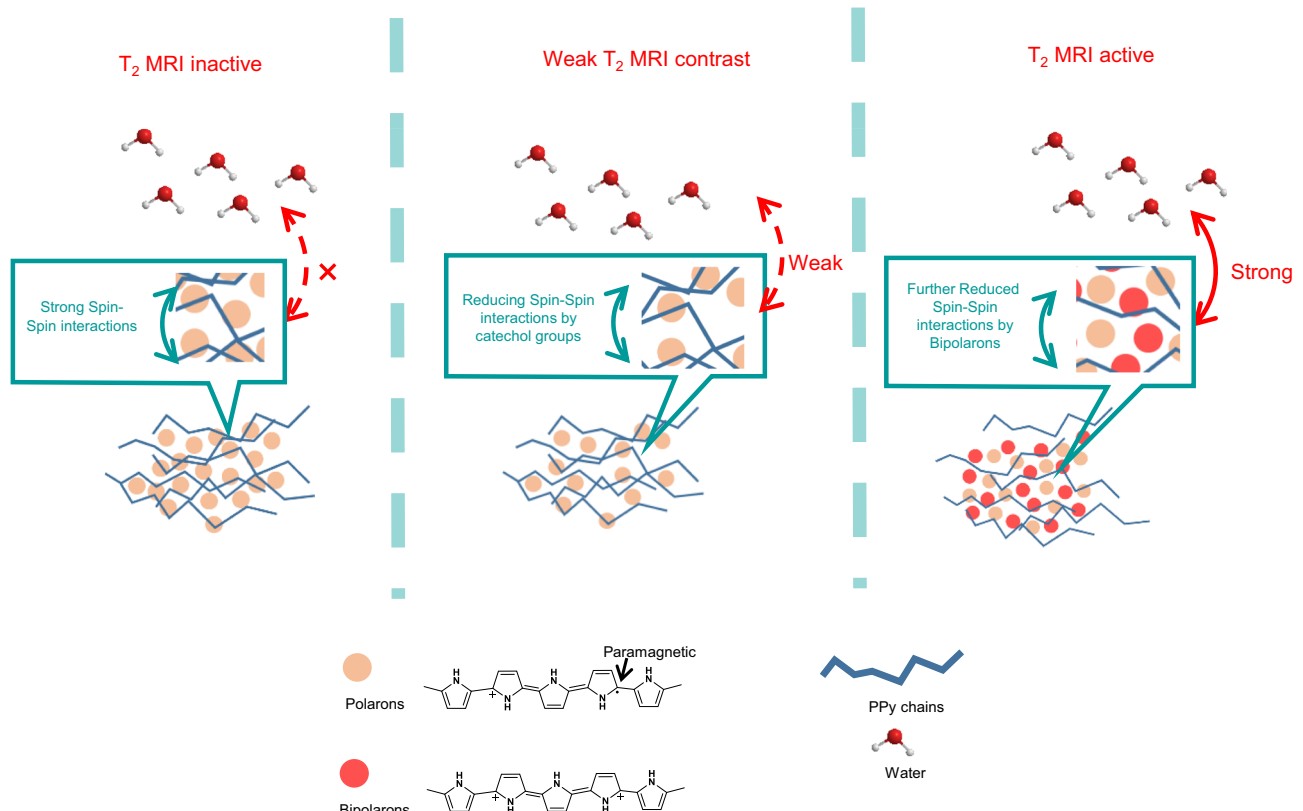

**Fig. 2 Schematic illustration of the "partially complexed" method for activating conjugated polymer nanoparticles in $T_2$ MRI.** For $T_2$ MRI inactive PPy nanoparticles, paramagnetic polarons are block from water protons by strong "spin–spin" interactions between polarons. For PPy nanoparticles with weak $T_2$ MRI contrast, above interactions are moderately reduced by catechol derivatives used in "partially complexed" method. For $T_2$ MRI active PPy nanoparticles, bipolarons further increase the interactions between paramagnetic polarons and water protons.

bipolarons. As an optical imaging technique, PAI possesses high sensitivity and temporal/spatial resolution[3]. Moreover, with centimeter penetration depths, minimized tissue absorption, and reduced background signals, NIR-II PAI contrast of catechol–PPy nanoparticles holds great promise for deep tissue imaging compared with that in NIR-I region[30,31].

After investigations about imaging abilities, a possible mechanism for catechol derivatives regulating the relative content of bipolarons is also studied by correlating $R$-values and the electron- donating/-withdrawing effect of the substituted groups in catechol derivatives. Since bipolarons are coupled by two polarons[27,32,33], generating sufficient polarons and regulating the bipolaron-formation process are two key factors for increasing the relative content of bipolarons vs polarons. First, sufficient oxidative agent is the premise for oxidizing neutral PPy into sufficient polarons. To validate that, we used 0.1 DA–PPy and 2.3 DA–PPy nanoparticles as examples. When ferric ions are fully complexed with dopamine (2.3DA–PPy), the relative intensity bipolaron band (above 1000 nm) in the absorption spectrum is reduced compared with the polaron band (around 700 nm) (Fig. 1e). Also, the shortening effect to $T_2$ relaxation time of 2.3 DA–PPy nanoparticles is very weak compared with that of 0.1 DA–PPy (Fig. 1d).

Then, the bipolaron formation is regulated by the radical trapping ability of phenol groups on the catechol derivatives, an ability that is changed by substituted groups[34]. Therefore, catechol–PPy nanoparticles with different substituted groups exhibit different $R$-values. When substituted with electron-donating groups (e.g., dopamine), the catechol derivatives tend to exhibit a radical scavenger effect that can directly terminate the polarons. When substituted with electron-withdrawing groups

(e.g., DHBA), the catechol derivatives tend to trap instead of terminate polarons[34] so that the catechol derivatives regulate the bipolaron-formation process. We speculate that the radical trapping ability contributes to increasing the probability of polarons forming bipolarons. In addition, for catechol derivatives with strong electron- withdrawing groups (e.g., DHBN with a cyano group), the radical trapping ability of their phenol groups is very weak[34] so that they only produce a weak regulating effect on bipolaron formation. The reduction in polaron content of DHBN nanoparticles can be attributed to the coordination between $Fe^{3+}$ and cyano groups. This coordination hinders $Fe^{3+}$ generating sufficient polarons. Therefore, the maximum $R$-value lies in the DHBA group.

Also, it should be noted that both a radical scavenging and trapping process exist for catechol derivatives. Therefore, part of polarons can be directly reduced by catechol derivatives. Reduced polarons exhibit imine quinonoid structure (-N=). The variation of imine quinonoid structure can be estimated by analyzing the XPS peak of PPy nanoparticles. The fractions of imine quinonoid structure can be estimated by fitting the N 1s peaks (details are shown in the supplementary information) in the XPS spectra pristine and catechol–PPy nanoparticles. For DA/CA/DHBA–PPy nanoparticles with stronger $T_2$ MRI contrasts ($T_2$ relaxation times in Table 1), significant increase of the reduced polarons (imine quinonoid structure) can also be observed in these nanoparticles, as well as decrease of polaron content (Supplementary Fig. 4). This phenomenon reveals that polaron reduction is inevitable for catechol derivatives regulating the relative amount of bipolarons vs polarons in PPy nano-particles. Also, the fractions of reduced polarons exhibit little variation between pristine and DHBN–PPy nanoparticles

(Supplementary Fig. 4). This result further supports the above speculations that DHBN has very weak ability for trapping polaron radicals.

Since catechol derivatives can both terminate polarons and induce formation of bipolarons. Besides, it is worth noting that the variation of $R$-values appears to be moderate in comparison with the polaron content (Table 1). This phenomenom may be also attributed to part of polarons reduced by catechol derivatives. Hence, the $R$-values can hardly vary from two polarons into one bipolaron. Although the detailed polymerization process remains to be studied, the relationship between substitutional groups and $T_2$ MRI contrast of catechol–PPy nanoparticles proves that the radical trapping ability is the key factor in regulating bipolaron formation.

Since bipolaron amount is regulated by the radical trapping process in our "partially complexed" method, this method can also be used to regulate the bipolaron amount in other conjugated polymers prepared with radical polymerization. Herein we use PEDOT and PANI nanoparticles to characterize the general applicability of the "partially complexed" method. PEDOT shares nearly the same polymerization process and similar polaron/bipolaron structures (Supplementary Fig. 5a) with PPy[35]. Therefore, the partially complexed method works for elevating the bipolaron content in all four kinds of catechol–PEDOT (Fig. 3a) nanoparticles because they all exhibited the very broad bipolaron band above 1000 nm in their absorption spectra (Fig. 3b). Also, the relative amount of bipolarons reaches the maximum in DHBA–PEDOT nanoparticles. This is because the polaron band (ranging from 700 to 900 nm) of DHBA–PEDOT nanoparticles is the least obvious among five kinds of PEDOT nanoparticles.

For PANI (Fig. 3c), the "partially complexed" method was further modified because the polymerization process of aniline is different from pyrrole and EDOT. During the preparation of PANI, we introduce a "pre-initiation" step, in which free ferric ions and ferric–catechol-derivative complex are successively added. This is because aniline monomers need to be conjugated to reduce the oxidative potential first and then chain radicals can be generated for polymerization[36]. This process can be interfered by catechol derivatives. Hence, free ferric ions and the ferric–catechol complex should be added separately to generate sufficient polarons. Among the four types of catechol–PANI nanoparticles prepared with this modified "partially complexed" method, only CA–PANI appears in the bipolaron band above 1000 nm (Fig. 3d). Such results are different from catechol–PEDOT/PANI nanoparticles and can be attributed to different structures of polarons and bipolarons (Supplementary Fig. 5b). The above results about PEDOT and PANI nanoparticles prove that this "partially complexed" method using catechol derivatives can increase the bipolaron content in conjugated polymers with similar polymerization process. Also, the different bipolaron content between catechol–PANIs. Since the molecular structure and reactivity of each polarons is varied, the chemical structure of catechol derivatives should be changed accordingly.

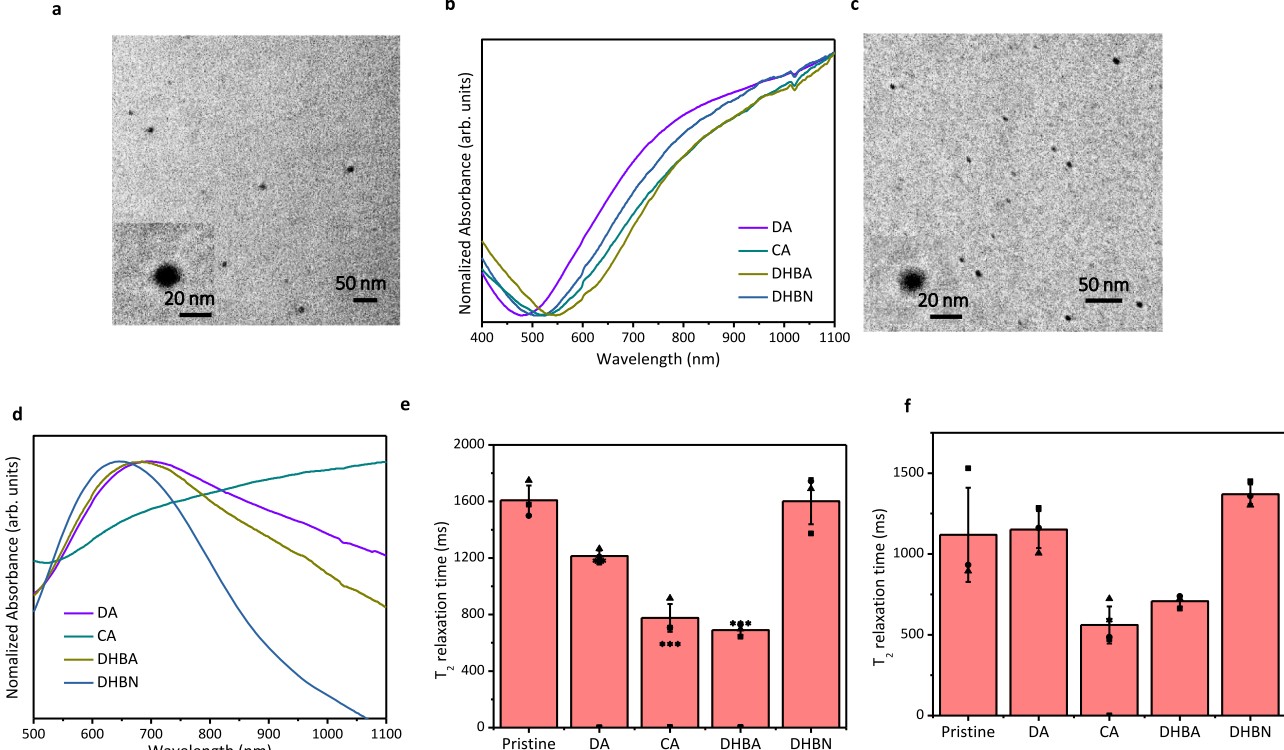

**Fig. 3 General applicability of the "partially complexed" method. a** Representative TEM images of catechol–PEDOT nanoparticles. Experiments were performed three times with similar results. **b** UV–Vis–NIR absorption spectra of pristine catechol–PEDOT nanoparticles. The discontinuous absorbance around 850 nm is attributed to the interchange of optical grating. **c** Representative TEM images of catechol–PANI nanoparticles. Experiments were performed three times with similar results. **d** UV–Vis–NIR absorption spectra of pristine catechol–PANI nanoparticles. The discontinuous absorbance around 850 nm is attributed to the interchange of optical grating. **e, f** $T_2$ relaxation times of PEDOT (**e**) and PANI (**f**) nanoparticles in 2 mg/mL, measured in aqueous solutions. Error bars in **e, f** represent mean ± s.d., with $n = 3$ independent experiments. Statistical significance for comparison of pristine PEDOT/PANI and catechol–PEDOT/PANI nanoparticles was determined by two-tailed Student's $t$-test (*$p < 0.05$, **$p < 0.01$, and ***$p < 0.001$). For PEDOT nanoparticles, $p_{DA} = 3.63E-03$, $p_{CA} = 5.51E-04$, $p_{DHBA} = 1.35E-04$, $p_{DHBN} = 0.958$. For PANI nanoparticles, $p_{DA} = 0.868$, $p_{CA} = 0.0367$, $p_{DHBA} = 0.0721$, $p_{DHBN} = 0.217$.

Next, we investigated the correlation between increased bipolaron content and $T_2$ MRI contrast in PEDOT and PANI nanoparticles. As expected, the $T_2$ relaxation times were still shortened in the catechol–PANI/PEDOT nanoparticles with increased bipolaron content (Fig. 3e, f). Those results revealed that bipolaron content is a key parameter for $T_2$ MRI contrast. It can either activate or enhance the $T_2$ MRI contrast of conjugated polymers. Moreover, the different $T_2$ relaxation times of PANI and PEDOT nanoparticles revealed that the MRI properties of conjugated polymers are also impacted by molecular structures. Therefore, the $T_2$ MRI contrast has great potential for further improvement by elaborating their molecular structures.

Furthermore, we also evaluated the feasibility of as-synthesized conjugated polymer nanoparticles for in vivo imaging. DHBA–PPy was chosen because it exhibits the highest $r_2$ relaxivity of polarons and NIR-II PAI contrast among all four groups of catechol–PPy nanoparticles. To alleviate biosafety concerns, as-synthesized nanoparticles should remain stable in physiological medium. Therefore, DHBA–PPy was first coated with DSPE–PEG through hydrophobic interactions. DHBA–PPy@DSPE–PEG nanoparticles were less than 100 nm in their dry state (Supplementary Fig. 6a). Also, a small proportion of DHBA–PPy aggregates can be observed in Supplementary Fig. 6b. Both the hydrophilicity of PEG chains and these aggregates increases the average hydrodynamic sizes of DHBA–PPy@DSPE–PEG nanoparticles (Supplementary Fig. 6c). After incubating DHBA–PPy@DSPE–PEG nanoparticles in complete DMEM culture medium for 7 days under sterilized environment at room temperature, the absorption spectrum as well as the hydrodynamic size of the nanoparticles remained similar compared with those in phosphate buffer saline (Supplementary Fig. 6c, d). At the cellular level, the nanoparticles are biocompatible for both L02 human liver cells and L929 mouse fibroblasts (Supplementary Fig. 7a). For the evaluation of in vivo toxicity, the biosafety of DHBA–PPy nanoparticles was evaluated using BALB/c mice. After 7 days, blood-biochemistry indexes and blood-panel analysis of the conjugated polymer nanoparticle (CPN) group remained in the normal ranges. This result indicated that the liver functions are not disrupted by DHBA–PPy@DSPE–PEG nanoparticles. Also, injected nanoparticles did not cause disorders to major organs according to the hematoxylin–eosin staining (H&E) images of the CPN group and control group (injected with PBS) (Supplementary Fig. 7b, c and Supplementary Table 2). The above results reveal that the injected nanoparticles are biosafe for both in vitro and in vivo applications.

In vivo $T_2$-weighed MRI and NIR-II PAI were carried out on BALB/c nude mice intravenously administered with DHBA–PPy@DSPE–PEG nanoparticles. $T_2$-weighed MRI was performed using a low-field (0.5 T) MRI system. After administering DHBA–PPy@DSPE–PEG nanoparticles to nude mice for 2 h, a darkened region in the liver was clearly observed (dotted circle in Fig. 4a). The signal-to-noise (S/N) ratio of this region obviously decreases from 5.5 to 2.8 (Fig. 4b). Besides, the S/N ratio increases to 3.5 after a further 3 h (Fig. 4b). The variation of S/N ratio with time suggests that such nanoparticles above 10 nm in diameter were mainly captured and detoxified by the reticuloendothelial system[37]. Also, such variation proved that the DHBA–PPy@DSPE–PEG nanparticles can keep "$T_2$ MRI active" state for at least 5 h. While usually small molecules of nitroxide radicals can only remain stable for several minutes. Current approaches to extend their in vivo stability to hours order involve complicated chemical modification processes. However, such stability can be easily reached in conjugated polymers with our simple but effective "partially complexed" method.

Besides, the enrichment of injected nanoparticles in the liver can also be visualized using NIR-II PAI. Before administration, the liver region exhibited no background signal at 1280 nm. After administering for 2 h, the photoacoustic signal produced by the nanoparticles

could also be detected in the liver (Figs. 4c and d). Moreover, the biodistribution of the nanoparticles could be clearly visualized after 3D reconstruction of PA images (Fig. 4e). For noninvasive imaging, PAI in the NIR-II region can complement $T_2$-weighed MRI with its high temporal/spatial resolution. These results revealed the feasibility of utilizing the physical properties of polarons and bipolarons in CPNs for in vivo imaging.

To evaluate the applications of our conjugated polymer-based $T_2$ MRI agents, we used $Fe_3O_4$ nanoparticles as a standard reference for in vivo imaging. As expected, the $T_2$ MRI contrast as well as the $r_2$ relaxivity of conjugated polymers are fairly weak in comparison with those of $Fe_3O_4$ nanoparticles (Fig. 4b and Supplementary Fig. 8). This is because the concentration of unpaired electrons in $Fe_3O_4$ nanoparticles is about 4–5 orders higher than that of PPy nanoparticles. Hence, finding other possible methods to increase the polaron content and keep the "MRI-active" status of polarons unchanged is necessary for developing conjugated polymer-based $T_2$ MRI agents with $Fe_3O_4$ comparable imaging contrasts.

In conclusion, we developed a "partially complexed" method that uses catechol derivatives to prepare $T_2$ MRI-active conjugated polymers. $T_2$ MRI contrasts of conjugated polymers are related to two factors: catechol derivatives for reducing the "spin–pin" interactions between polarons and increased bipolarons for enhancing the interactions between polarons vs water protons. Subsequently, we pointed that the radical trapping ability of catechol derivatives regulates the formation of bipolarons. Therefore, this "partially complexed" method is generally applicable in similar conjugated polymers to activate/increase the $T_2$ MRI contrasts of conjugated polymers.

Furthermore, this research introduces conjugated polymers as a class of nonmetallic $T_2$ MRI agents. Conjugated polymer-based agents not only exhibit satisfying in vivo stability but might be a reference for the development of a series of nonmetallic $T_2$ MRI agents. This is because the oxidized forms are a common phenomenon in conjugated polymers of various molecular structures. We believe that conjugated polymers can open a field in nonmetallic MRI agents by overcoming the problem of unpaired electron concentration.

## Methods

**Ethical statement**. Animal experiments comply with the National Institutes of Health Guide for the Care and Use of Laboratory Animals. All animal care and handling procedures were approved by the ethics committee of Fudan University.

**Synthesis of conjugated polymer nanoparticles**. The reaction solution was prepared by dissolving DeTAB and monomers in 4 mL of 1 M HCl. The oxidation solution was prepared by dissolving ferric chloride hexahydrate and catechol derivatives in 1 mL of 1 M HCl. The oxidation solution was added into the reaction solution drop by drop. For preparing PPy nanoparticles, the concentration of DeTAB was 0.6 M, pyrrole monomer 0.749 mmol, and ferric chloride hexahydrate 1.73 mmol. The reaction was processed in an ice bath for 6 h. For PEDOT nanoparticles, the concentration of DeTAB was 0.21 M, 3,4-ethydioxythiophene monomer 0.704 mmol, and ferric chloride hexahydrate 1.48 mmol. The reaction was kept at room temperature for 48 h. For PANI nanoparticles, the concentration of DeTAB was 0.4 M, aniline monomer 1.07 mmol, and ferric chloride hexahydrate 2.14 mmol. Free ferric ions were added into the reaction solution first. This process was named as "pre-initiation" in this research. After 1 h, the ferric–catechol complex was added into the reaction solution. The reaction was kept at room temperature for 48 h. After the reaction was completed, the nanoparticles were centrifuged and washed with ethanol several times, and then the nanoparticles (1 mg) coated with 10 mg of DSPE–PEG by sonication for 10 min. Excessive DSPE–PEG was removed by centrifugation. DHBA–PPy@DSPE–PEG nanoparticles were subsequently washed by deionized water for 2–3 times by centrifugation at 10000 g for 10 min.

**Physical characterization of as-synthesized conjugated polymer nanoparticles**. Transmission electron microscope (TEM) images were acquired on a FEI TEM (Tecnai G2 20 TWIN), with an acceleration voltage of 200 kV. UV–Vis–NIR absorption spectra of different CPNs coated with DSPE–PEG were

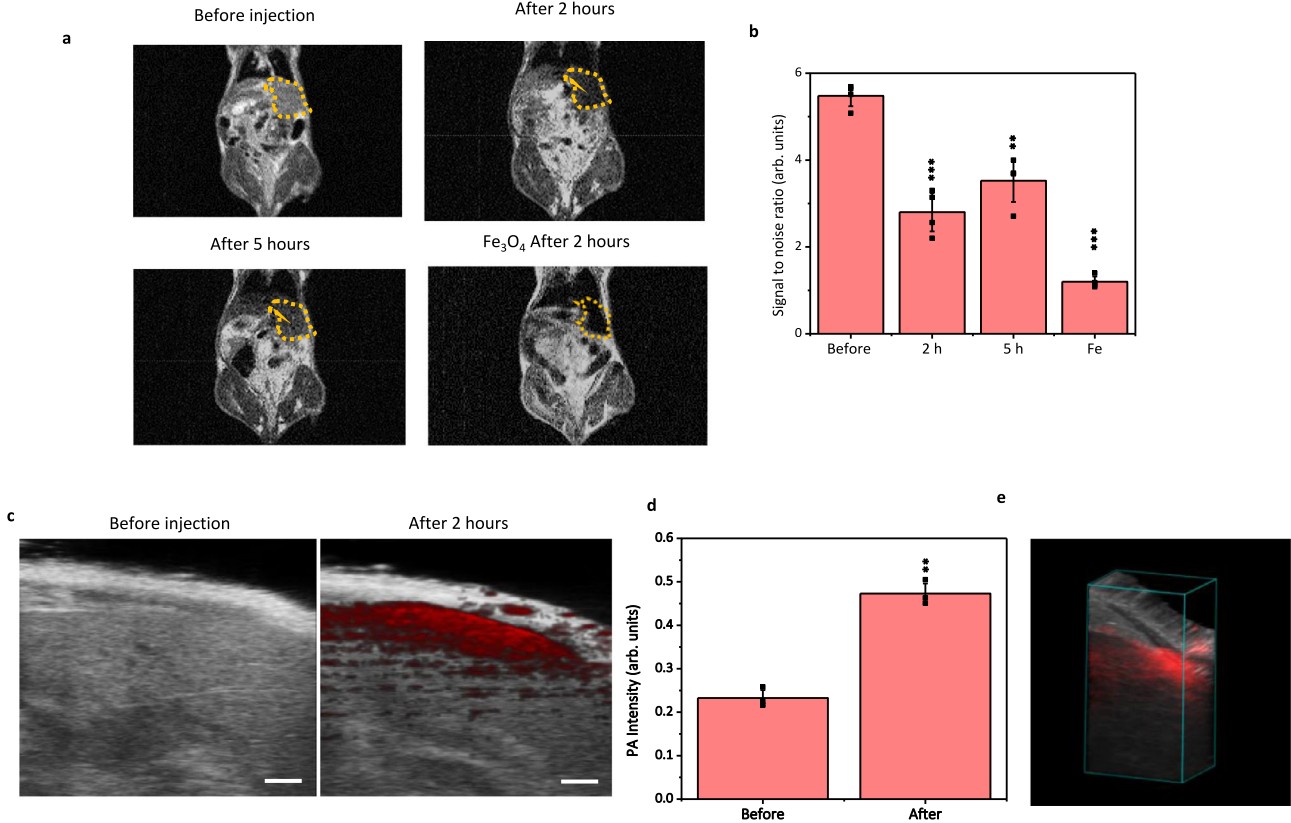

**Fig. 4 Feasibility of DHBA–PPy@DSPE–PEG nanoparticles for in vivo imaging. a** $T_2$-weighed MRI images of BALB/c nude mice before, after 2 h, after 5 h of injection of DHBA–PPy@DSPE–PEG nanoparticles, and after 2 h of injection of $Fe_3O_4$ nanoparticles. **b** The variation of signal/noise ratio of the dotted circled region in the liver. Error bars in b represent mean ± s.d., with $n = 4$ independent experiments. Statistical significance for comparison of before injection, after 2 h, after 5 h, and $Fe_3O_4$ groups was determined by two-tailed Student's $t$-test (*$p < 0.05$, **$p < 0.01$, and ***$p < 0.001$). $p_{2h} = 4.05E\text{-}05$, $p_{5h} = 3.57E\text{-}04$, $p_{Fe} = 6.37E\text{-}08$. **c** Photoacoustic imaging acquired at 1280 nm of the liver region before and after injection of DHBA–PPy@DSPE–PEG nanoparticles. The scale bar is 1 mm. Experiments were performed three times with similar results. **d** The variation of photoacoustic intensity in the liver. Error bars in b represent mean±s.d., with $n = 3$ independent experiments. Statistical significance for comparison of before injection and after 2 h was determined by two-tailed Student's $t$-test (**$p = 1.46E\text{-}04 < 0.01$). **e** 3D reconstruction (with 1mm step) of photoacoustic images acquired at 1280 nm of the liver region after injection of DHBA–PPy@DSPE–PEG nanoparticles.

acquired on a Perkin-Elmer spectrometer (Lambda 750). Each absorption spectrum was the average of 3 scans. The discontinuous signal around 850 nm was caused by the interchange of excitation lamps. X-ray diffraction (XRD) measurements were carried out at room temperature on a Bruker diffractometer (D2 PHASER) using Cu Kα radiation (wavelength = 1.5406 Å). The dynamic light scattering was performed on a Malvern particle-size analyzer (ZS-90). ESR spectra of PPy dry powders were recorded on a Bruker EPR spectrometer (E500-10/12). The microwave frequency was 9.37 GHz, attenuator 14.0 dB, modulation frequency 100.0 kHz, and the modulation amplitude 5.0 G. The spinfit function in Xepr software (version 2.6) was used to determine the spin density of PPy samples after a factory calibration of the cavity-response function. The uncertainty of the spin concentration was estimated to be within ~10%. Raman spectroscopy of PPy dry powders was performed on a Jobin Yvon Horiba (Xplora) connected to a CCD detector. A He–Ne laser, 785 nm, was used for excitation in conjunction with a 600-line grating and a 50× lens. Each spectrum was acquired with 1% laser power, 5 s of accumulation time, and 50-times accumulation. Baseline corrections were performed using LabSpec6 software. The fluorescence background was removed from the spectrum with a polynomial fourth-order baseline algorithm. FTIR spectra were acquired on a Thermofisher FTIR spectrometer (Nicolet 6700). $T_2$ relaxation times were acquired on a 0.5 T Niumag micro MR analyzing system (PQ001-20-015V). Data analysis of absorption/FTIR/XRD spectra and statistical analysis were performed using Origin 8.5 and Microsoft Excel 2019. Molecular structures were performed with ChemDraw Professional 16.0.

**X-ray photoelectron spectroscopy (XPS) analysis**. XPS spectra were acquired using a Perkin-Elmer XPS System (PHI 5000 C ESCA) with a Mg Kα achromatic X-ray source (1253.6 eV) at a power of 250 W, 14 kV. The total pressure in the main vacuum chamber during analysis was typically between $10^{-7}$ and $10^{-8}$ Pa.

Survey spectra were acquired at a pass energy of 93.9 eV and a step size of 1 eV. To obtain more detailed information about chemical structure, oxidation states, etc., narrow scans were recorded from individual peaks at 93.9-eV pass energy and 0.2-eV step size. The synthesized nanoparticles were washed with ethanol and dried under vacuum. Samples for XPS measurement were prepared by compressing about 15 mg of dried nanoparticles into conductive carbon tapes. Also, the sample surfaces were also grounded with the sample mount by conductive carbon tapes for minimizing potential charging effect.

Data processing was performed using CASAXPS processing software version 2.3.24. All elements present were identified from survey spectra. The atomic concentrations of the detected elements were calculated using integral peak intensities and the sensitivity factors supplied by the manufacturer. Binding energies were referenced to the C 1$s$ peak at 284.9 eV (combination of aromatic and aliphatic hydrocarbon at 284.7 eV and 285.0 eV, respectively) with an uncertainty of ±0.2 eV[38]. For fitting the N 1$s$ peak, we use a slightly modified Universal Polymer Tougaard background and pseudo-Voigt functions (LA line shapes in CasaXPS) for the component peaks[38,39]. Slightly asymmetric should be accounted for aromatic structures. This is because a proportion of the photon energy may contribute to atomic oscillations associated with the bonds within a molecule and is not available to the photoelectron as kinetic energy[39]. The consequence of these alternative energy-transfer mechanisms is that the XPS peaks exhibit asymmetry. The full width at half-maximum (FWHM) is constrained to a narrow range of 1.4–1.8 eV. We use four components at 398.5 ± 0.2 eV (quinonoid imine, -N =), 399.7 ± 0.2 eV (benzenoid amine, -NH-), and two peaks at about 401.2 ± 0.2 eV and 402.7 ± 0.2 eV for purposes of tracking the peak-shape changes[40]. The high BE tail above 400 eV can be contributed by various species, including protonated imine/amine, surface-oxidation products, interchain H-bonding, and the imine nitrogen satellite[40,41]. Thus, no attempt to assign specific species to each peak will be made. The error analysis of peak-fitting results is

performed using Monte-Carlo simulations in CASAXPS. The standard deviation of the peak area based on peak-fitting results is about 1–5%. It should be noted that the standard deviations are correlated to the constraints of peak-fitting parameters. Tight constraints can reduce the standard deviations.

**Cellular experiments**. Cell toxicity was evaluated using human liver L02 and L929 fibroblast cells. Both cell lines were purchased from the Stem Cell Bank, Chinese Academy of Sciences. L929 fibroblast cells were cultured in DMEM culture medium containing 10% fetal bovine serum and 1% penicillin and streptomycin. Human L02 liver cells were cultured in RPMI-1640 culture medium containing 20% fetal bovine serum and 1% penicillin and streptomycin. Both cell lines were cultured at 37 °C in a 5% $CO_2$ atmosphere. About $10^4$ cells/well were seeded in a 96-well plate for 24 h ($n = 5$). Then, the complete culture medium was replaced by serum-free culture medium containing DHBA–PPy@DSPE–PEG nanoparticles. The nanoparticles and cells were co-incubated for a further 24 h. Cells treated with serum-free culture medium without nanoparticles were taken as control group. Cell viability was tested using the CCK-8 assay on a Biotech microplate reader (800 TS).

**Animal experiments**. Female BALB/c and BALB/c nude mice (6 weeks old) were purchased from the Shanghai SLRC Laboratory Animal Centre. Housing conditions were kept 22–26 °C in temperature, 40–60% in humidity, and 12h light–dark cycle. To evaluate the toxicity of nanoparticles in vivo, 3 healthy BALB/c mice were injected with a 200-μL suspension of DHBA–PPy@DSPE–PEG nanoparticles (0.2 mg/mL) and another 3 healthy mice were injected with PBS (pH 7.4) as the control group. After 7 days, the injected mice were sacrificed, blood samples collected, and major organs harvested, fixed in 4% formalin, and prepared as paraffin-embedded sections for H&E staining. Standard serum-biochemistry levels were tested, including alanine aminotransferase (ALT), alkaline phosphatase (ALP), aspartate aminotransferase (AST), albumin/globulin ratio (A/G), and urea. Complete blood-panel analysis was tested, including white blood cells (WBC), red blood cells (RBC), hemoglobin (HGB), hematocrit (HCT), mean corpuscular volume (MCV), mean corpuscular hemoglobin (MCH), mean corpuscular hemoglobin concentration (MCHC), and platelets (PLT).

For in vivo imaging, BALB/c nude mice were injected with 200 μL of 0.5 mg/mL DHBA–PPy@DSPE–PEG nanoparticles. After 2 h, nude mice were anesthetized using 2% isoflurane in oxygen. PA images were acquired at an excitation wavelength of 1280 nm on a Fujifilm VisualSonics Photoacoustic Imaging system (Vevo Lazer-X) equipped with a linear-array transducer (256 elements) to detect US and PA signals, and a tunable Nd:YAG laser system (680–970 nm/1200–2000 nm, ≥20-Hz repetition rate, and ≥40-mJ pulse peak energy) was used to trigger the system acquisition. An acquisition rate of 20 frames per second was used for all experiments. Photoacoustic images were processed using a Fujifilm VevoLab software version 3.1.0. $T_2$-weighed MRI images were acquired on a 0.5 T Niumag nuclear magnetic resonance imaging system (MesoMR-60-H-I), using a standard multislice spin-echo pulse sequence, with an echo time (TE) of 50 ms, a repetition time (TR) of 1300 ms, and a slice thickness of 1.5 mm. To evaluate the MRI contrast of the liver region, we defined regions of interest (ROIs) around the left lobes of the liver on the in vivo MR images. The signal-to-noise ratio was analyzed using ImageJ software version 1.52a.

**Reporting summary**. Further information on research design is available in the Nature Research Reporting Summary linked to this article.

## Data availability

All data supporting the findings from this study are provided in the paper and its supplementary information. All data are also available from the corresponding author upon request. Source data are available for Figs. 1b–g, 3c-f, 4b, and c and Supplementary Figs. 1a–d, 2a, b, 3a, b, 4a–g, 6c, d, 7a, b, and 8 in the associated Source Data file. Source data are provided with this paper.

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

## Acknowledgements

This work was supported by the Key Program of National Natural Science Foundation of China (21935002 to Z. S.) and Ministry of Science and Technology of the People's Republic of China (MOST) (2016YFA0203301 to Z. S.). Also, many thanks for Dr. Chen and Dr. Endler in Shanghai BIOMED Science Technology Co., Ltd. for reading of and corrections in the paper, Prof. Dai Weilin in the Department of Chemistry, Fudan University for giving suggestions about processing the XPS data, and Dr. Zhang in Shanghai Institute of Organic Chemistry, Chinese Academy of Sciences for ESR measurement.

## Author contributions

Q.L. synthesized the nanoparticles, conducted the characterizations, and animal experiments, and was primarily responsible for data collection. Z.S., Q.L., and Y.Y. designed the project, analyzed the results, and prepared the paper, figures, and supplementary information.

## Competing interests

The authors declare no competing interests.

## Additional information

**Peer-review information** *Nature Communications* thanks the anonymous reviewers for their contribution to the peer review of this work. Peer-reviewer reports are available.

