## [Peer Review File · Nature Communications]

REVIEWER COMMENTS

Reviewer #1 (Remarks to the Author):

This submission reports the development of conjugated polymer nanoparticles for magnetic resonance imaging (MRI). The authors chemically modulated the oxidized forms of the polymer and explored their magnetic properties for MRI. While the optical properties of conjugated polymers have been extensively studied, their magnetic properties are much less investigated. This aspect of the work is interesting and novel, but some of the results in the work are questionable and the presentation is very confusing.

The authors admitted the presence of "negligible quantities of inorganic compounds in the nanoparticles", but no data was shown. The r_2 relaxivity of the nanoparticles is very small, according to Table 1, the r_2 of PPY NPs is less than 1 mL/(mgs). In comparison, the r_2 relaxivity of iron oxide nanoparticles is typically around a few thousand mL/(mgs) (or a few hundred mM⁻¹s⁻¹). So the possibility of the contribution of "negligible quantities of inorganic compounds" (less than ug) to this observed low relaxivity is unreasonably excluded.

Table 1 values are inconsistent: based on the polaron content (/mg) and r_2 relaxivity of PPy nanoparticles (mL/mgs), the value of r_2 relaxivity of polarons is off by three orders of magnitude, 1000 (I converted the number of polaron to mole number –for example: 4.51×10^{14} /mg / 6.02×10^{23} /mol = 7.49×10^{-10} mol/mg) then converted r_2 of PPy NPs to that of polaron: for example, 0.467 mL/mg.s / 7.49×10^{-10} mol/mg = $623,000$ L/mol.s), which would be close to that of SPIO relaxivity!

Fig.1 a: if polarons are paramagnetic and bipolarons are diamagnetic, then should the polaron CPN be T2-MRI active instead of bipolaron CPNs be T2-MRI active? Throughout the paper, sometimes it implies polaron responsible for T2 MRI activity and the other places attribute it to bipolarons.

The MR imaging is so cursory, and the contrast is weak and not that obvious from the image. The authors should at least compare it to a standard reference in a phantom study.

On the calculation of relative amount of bipolarons vs polarons using the Raman spectra in Fig. 1d: I could not see the peak 1086 nm. How did the authors assign them to each species?

Table 1: the authors speculate that catechol derivatives modulate the polymerization process and thus the ratio of polaron and bipolaron. But how did that modulate the total number of polaron per mg mass? It seems the variation of the number of polaron is much bigger than the ratio.

If polaron is paramagnetic and responsible for the high r_2 relaxivity, then why pristine PPy NPs have the most polaron content, but the smallest r_2 relaxivity (Table 1)? On the other hand, it may be related to the absence of iron metal in the synthesis.

On the biosafety of the nanoparticles: the blood panel should also be performed at the 12 hours postinjection; The authors should show the biodistribution and clearance of the particles after introduction.

Fig. 2d: no circles are drawn to show the liver.

Page 8: line 3 from the end of the paragraph: the normal range (Fig. 2a) should be Fig. 2b; the last sentence, (Fig. 2b and Table S1) should be Fig. 2c and Table S1.

Fig.s4b misses the plot for partially complexed group.

Fig.s6: it seems there are two types of nanoparticles in the samples: one shows dark contrast and the others show light contrast. Do the authors know which is the polymer nanoparticles? Could the other one be the inorganic nanoparticles with the iron metal?

Reviewer #2 (Remarks to the Author):

In this manuscript, the authors claimed to develop a series of T2 -weighted MRI active conjugated polymers (catechol-PPy, catechol-PEDOT or catechol-PANI nanoparticles) by using catechol derivatives to regulate the relative ratio of bipolarons to polarons. Although the concept is quite interesting, data presented here are not sufficient to fully support the proposed mechanism. And the proposed mechanism is ambiguous that lacks detailed explanation. Also, the imaging properties of these nanoparticles are not properly validated. Thus, a major revision is needed before its publication.

1. The authors should provide the specific chemical structures or at least the proposed chemical structures of catechol-PPy, catechol-PEDOT and catechol-PANI. Also, corresponding chemical characterization (e.g. ¹H NMR and gel permeation chromatography) should be performed to validate the synthesis of these compounds. In particular, please provide data to confirm the existence and the structure of the so called "oxidized species" in the catechol-polymer complex.
2. Catechols as a type of phenols have been reported as a family of T2-weighted MRI contrast agents (e.g. Chemistry, 2018, 24, 1259-1263). In this manuscript, it is confusing and difficult to confirm in the catechol-polymer complex that whether it is the polymer or the catechol itself to play the major role as the MRI contrast agent. The authors should provide T2 -weighted MRI signals from these catechols as control.
3. Given the central topic of this work is about metal-free MRI contrast agents, the authors should mention in the introduction the description about phenols (catechols included) for chemical exchange saturation transfer (CEST) MRI.
4. T2 -weighted MRI signals and images from catechol-PPy, catechol-PEDOT and catechol-PANI nanoparticles should be provided in the manuscript. Also, NIR-II PA signals and images of these nanoparticles should be provided.
5. The relative T2 MRI signal intensity of catechol-PPy, catechol-PEDOT or catechol-PANI nanoparticles should be benchmarked with the broadly reported contrast agents such as superparamagnetic iron-oxide NPs (SPIO-NPs) to show their potential in MRI imaging.
6. Quantification and statistical analysis of Fig. 2d and 2e should be performed.

Reviewer #3 (Remarks to the Author):

In this work, the authors reported T2-weighted MRI active polypyrrole nanoparticles and correlated their paramagnetism with the formation of bipolaron. Although using CPN as MRI contrast agent is interesting, however, the manuscript could hardly meet the high criteria for publication in nature communication. Below are my concerns and suggestions.

1. The reason for the r2 relaxivity of DHBN-PPy NPs to be much smaller than the other catechol-PPy NPs should be clearly explained.
2. Considering the R value of DHBA is moderately higher than the other catechol-PPy NPs, the reason for its r2 relaxivity of polaron to be much higher than the others should be explained.
3. The author also fabricated PEDOT and PANI NPs and analyze their bipolarons content. Why their T2 relaxation time measurement result was not shown? Their MRI performance are important to support the bipolaron theory.
4. The UV-Vis-NIR absorption spectra of pristine PPy should be provided and taken into the discussion of bipolarons generation.
5. It is not clear why PVA-ppy shows much lower amount of bipolarons as compared with catechol-PPy.
6. The author should do a comparison of MRI imaging between PPy NPs and commercial MRI contrast.
7. The analysis of substitutional group effect on the formation of bipolar is very confusing. When the substitutional groups of catechol derivatives varied from electron donating to neutral and electron withdrawing, the radical scavenger effect decreased and thus polymerization process was less affected, leading to more polarons. The authors should clearly explain the moderate radical

scavenger effect.

8. Figure 1a is very confusing. The structures of synthesized catechol-PPy should be given and the interaction between catechol and polaron/bipolaron should be highlighted.

9. Figure 1b, the quality of TEM image is low. And the average size is obviously not 20 nm as described but 10-15 nm.

10. The calculation method for polaron content should be provided.

11. Many related literature should be cited, especially in the introduction part.

Reviewer #1:

General comments: This submission reports the development of conjugated polymer nanoparticles for magnetic resonance imaging (MRI). The authors chemically modulated the oxidized forms of the polymer and explored their magnetic properties for MRI. While the optical properties of conjugated polymers have been extensively studied, their magnetic properties are much less investigated. This aspect of the work is interesting and novel, but some of the results in the work are questionable and the presentation is very confusing.

Response R1: Thank you for taking the time to review our manuscript. According to your valuable suggestions, we have rearranged the Results and Discussion sections and carefully re-evaluated the impact of residual metal elements on the T₂ MRI contrast. Please read our point-by-point responses below. We believe the revised manuscript now addresses your important concerns.

Comment 1 (C1): The authors admitted the presence of “negligible quantities of inorganic compounds in the nanoparticles”, but no data was shown. The r_2 relaxivity of the nanoparticles is very small, according to Table 1, the r_2 of PPY NPs is less than 1 mL/(mgs). In comparison, the r_2 relaxivity of iron oxide nanoparticles is typically around a few thousand mL/(mgs) (or a few hundred $\text{mM}^{-1}\text{s}^{-1}$). So the possibility of the contribution of “negligible quantities of inorganic compounds” (less than μg) to this observed low relaxivity is unreasonably excluded.

Response R1-1: Many thanks for raising this important issue. The possible inorganic compounds are brought by the oxidant ferric chloride, which is used for preparing both pristine and catechol-PPy nanoparticles. To analyze these possible “inorganic compounds”, the decomposing curves of pristine and catechol-PPy nanoparticles in air were characterized by thermogravimetric analysis. For both pristine and catechol-PPy nanoparticles, the remaining decomposition products

may be formed by residual iron elements and the mass fractions of the decomposition products are about 1-2%. However, only catechol-PPy nanoparticles exhibit T₂ MRI contrast. This results corroborated the data that residual iron elements are not the determining factor for the T₂ MRI contrast of catechol-PPy nanoparticles.

Corresponding content in the revised manuscript:

Page 5, paragraph 1:

Response R1-1: The impact of residual iron elements brought on T₂ MRI contrast were firstly excluded by analyzing the iron content. Since ferric chloride was used as the oxidant to prepare both pristine and catechol-PPy nanoparticles, residual metal elements may form negligible inorganic compounds that contributes to the T₂ MRI contrast in PPy nanoparticles. Here, the level of residual metal elements was evaluated by thermal decomposition of PPy nanoparticles in air. This thermogravimetric analysis reveals that decomposition product (may be the oxides formed by residual iron elements) remains at a level of only about 1-2% for both pristine and catechol-PPy nanoparticles (Fig. S1a). Also, such a low content of metal elements is likely to form only negligible levels of T₂ MRI active inorganic compounds. Hence, the characteristic sharp peaks of inorganic compounds were not observed in the XRD patterns of catechol-PPy nanoparticles (Fig. S1b). Although the contents of residual iron elements were similar for pristine PPy and catechol-PPy nanoparticles, only catechol-PPy nanoparticles exhibited T₂ MRI contrast. Therefore, residual metal elements are not the determining factor for generating T₂ MRI contrast in catechol-PPy nanoparticles.

C2: Table 1 values are inconsistent: based on the polaron content (/mg) and r₂ relaxivity of PPy nanoparticles (mL/mgs), the value of r₂ relaxivity of polarons is off by three orders of magnitude, 1000 (I converted the number of polaron to mole number –for example: $4.51 \times 10^{14} / \text{mg} / 6.02 \times 10^{23} / \text{mol} = 7.49 \times 10^{-10} \text{ mol/mg}$) then converted r₂ of PPy NPs to that of polaron: for example, $0.467 \text{ mL/mg.s} / 7.49 \times 10^{-10} \text{ mol/mg}$

= 623,000 L/mol.s), which would be close to that of SPIO relaxivity!

Response R1-2: Thank you for pointing out this important issue. We made a mistake in our definition of the units of polaron r_2 relaxivity, which should be L/mmol·s; sorry about that but we are grateful that you pointed out this error. In this manuscript, we introduced the r_2 relaxivity of polarons to reflect the variation of polaron status. This parameter is not applicable for making comparisons with other T_2 MRI agents because the polaron contents and T_2 relaxation time were measured under different conditions. The polaron contents were measured in dry powders while the T_2 relaxation times were measured in different conditions. Because water strongly attenuated the excitation energy of electron spin spectroscopy, the polaron content of PPy nanoparticles in water is barely detectable. The polaron content in water may well be different from the values we measured in dry powders.

Corresponding content in the revised manuscript:

Page 7, caption for Table 1

Response R1-2: r_2 relaxivity of polarons were estimated from the spin densities, which were characterized in dry powders, while T_2 relaxation times of PPy nanoparticles were measured in aqueous solution. Therefore, r_2 relaxivity was only used to investigate the polaron status.

C3: Fig.1 a: if polarons are paramagnetic and bipolarons are diamagnetic, then should the polaron CPN be T_2 -MRI active instead of bipolaron CPNs be T_2 -MRI active? Throughout the paper, sometimes it implies polaron responsible for T_2 MRI activity and the other places attribute it to bipolarons.

Response R1-3: Thank you for raising this important point. Sorry for the previous confusing conclusions, therefore we have accordingly revised our manuscript to make our conclusions clearer. The T_2 MRI contrast is generated by paramagnetic species, which are polarons in the PPy nanoparticles. The role of bipolarons in

MRI contrast is to change the polaron status to increase significantly the MRI contrast of the catechol-PPy nanoparticles.

Corresponding content in the revised manuscript:

Page 6, paragraph 3

Response R1-3: The contrast of MRI agents is impacted by two factors, paramagnetic species content and their status.

Page 7, paragraph 1

Response R1-3: Therefore, the reduced linewidth in catechol-PPy nanoparticles confirmed that their T₂ MRI contrasts are also related to a varying polaron status.

Page 8, paragraph 1

Response R1-3: This observation revealed that variation in the relative content of bipolarons changed the status of the paramagnetic polarons. This process was responsible for activating catechol-PPy nanoparticles in T₂ MRI.

C4: The MR imaging is so cursory, and the contrast is weak and not that obvious from the image. The authors should at least compare it to a standard reference in a phantom study.

Response R1-4: It is a really helpful point for our paper. Accordingly, we chose superparamagnetic Fe₃O₄ nanoparticles synthesized by hydrothermal method as a standard reference for MRI properties of our conjugated polymer nanoparticles. The r₂ relaxivity of Fe₃O₄ nanoparticles is about 9.9 mL/mg·s, which is higher than our conjugated polymer agents as expected. This is because the fraction of paramagnetic species in conjugated polymers (about 10⁻⁹ mol/mg) is lower than that in Fe₃O₄ nanoparticles (about 10⁻⁵ mol/mg). While the concept of this research is introducing a new class of T₂ MRI agents rather than enhancing the imaging abilities of existing MRI agents. Further improving the MRI properties of conjugated polymers can certainly be reached in following researches by tuning their molecular structures.

Corresponding content in the revised manuscript:

Page 15, paragraph 2

Response R1-4: Furthermore, this research might be a reference for the development of a series of nonmetallic T₂ MRI agents that are not limited by particular molecular structures. This is because the oxidized forms are a common phenomenon in conjugated polymers of various molecular structures. Therefore, the imaging abilities of conjugated polymers, although not comparable with other newly developed inorganic T₂ MRI agents (Fig. S8), can certainly be improved by further tuning the structures of the conjugated polymers.

C5: On the calculation of relative amount of bipolarons vs polarons using the Raman spectra in Fig. 1d: I could not see the peak 1086 nm. How did the authors assign them to each species?

Response R1-5: Thanks for raising this question. The Raman band around 1086 cm⁻¹ of bipolarons is not obvious because it is overlapped with the Raman band of polarons around 1055 cm⁻¹. This bipolaron band is more obvious in other spectra of PPy nanoparticles as shown in Fig. S3. Also the relative amount of bipolarons vs polarons (noted as *R*-value) is a semi-quantified parameter. In this manuscript this ratio is used to represent the variation trend compared with r₂ relaxivity of polarons in the PPy nanoparticles rather than the actual ratio of these two oxidized species. This point has been added in the revision.

Corresponding content in the revised manuscript:

Page 8, paragraph 1:

Response R1-5: The *R*-value is used to represent the variation trend of the relative content of bipolarons vs polarons, rather than the actual ratio in PPy nanoparticles.

C6: Table 1: the authors speculate that catechol derivatives modulate the polymerization process and thus the ratio of polaron and bipolaron. But how did that modulate the total

number of polaron per mg mass? It seems the variation of the number of polaron is much bigger than the ratio.

Response R1-6: Many thanks for pointing out this issue. We have rewritten this part in revision. The interactions between polarons and catechol derivatives are divided into two process. The first one is radical scavenger process, in this process polarons are directly terminated through electron or hydrogen atom transfer process. Another is radical trapping process, in this process polarons are temporally trapped and the possibility for them coupling into bipolaron increased. Both processes can cause the variation of the total number of polaron per mg mass. Since both processes exist in the preparation of PPy. The ratio of bipolarons vs polarons cannot varied strictly according to two polarons into one bipolaron. Therefore, the variation of the polaron number seems bigger than the ratio.

Corresponding content in the revised manuscript:

Page 9, paragraph 3:

Response R1-6: Then, the bipolaron formation is regulated by the radical trapping ability of phenol groups on the catechol derivatives, an ability that is changed by substituted groups³³. Therefore, catechol-PPy nanoparticles with different substituted groups exhibit different *R*-values. When substituted with electron donating groups (e.g. dopamine), the catechol derivatives tend to exhibit a radical scavenger effect that can directly terminate the polarons. When substituted with electron withdrawing groups (e.g. DHBA), the catechol derivatives tend to trap instead of terminate polarons³³ so that the catechol derivatives regulate the bipolaron formation process. We speculate that the radical trapping ability contributes to increasing the probability of polarons forming bipolarons. In addition, for catechol derivatives with strong electron withdrawing groups (e.g. DHBN with a cyano group), the radical trapping ability of their phenol groups is very weak³³ so that they only produce a weak regulating effect on bipolaron formation. Therefore, the maximum *R*-value lies in the DHBA group. Also, it should be noted that as both a radical scavenging and trapping process exist for

catechol derivatives, the R values can hardly vary strictly from two polarons into one bipolaron. Hence, the variation of R -values appears to be moderate in comparison with the polaron content (Table 1). Although the detailed polymerization process remains to be studied, the relationship between substitutional groups and R -value proves that the radical trapping ability is the key factor in regulating bipolaron formation.

C7: If polaron is paramagnetic and responsible for the high r_2 relaxivity, then why pristine PPy NPs have the most polaron content, but the smallest r_2 relaxivity (Table 1)? On the other hand, it may be related to the absence of iron metal in the synthesis.

Response R1-7: Thanks for raising this question. As we mentioned in R3, besides polaron content, polaron status is also a key factor for generating T_2 MRI contrast in PPy nanoparticles. In catechol derivatives, the polaron status are changed by the relative content of bipolarons vs polarons so that catechol-PPy nanoparticles can generate T_2 MRI contrast. Also, both pristine and catechol-PPy nanoparticles were synthesized using ferric chloride as the oxidants. The iron elements is not the determining factor for T_2 MRI contrast in PPy nanoparticles, which has been discussed in R1.

Corresponding content in the revised manuscript:

Page 5, paragraph 1:

Response R1-7: Although the contents of residual iron elements are similar in pristine PPy and catechol-PPy nanoparticles, only catechol-PPy nanoparticles exhibit T_2 MRI contrast. Therefore, residual metal elements are not the determining factor for generating T_2 MRI contrast only in catechol-PPy nanoparticles.

Page 6, paragraph 2:

Response R1-7: The contrast of MRI agents are impacted by two factors, paramagnetic species content and their status.

Page 7, paragraph 1:

Response R1-7: Therefore, the reduced linewidth in catechol-PPy nanoparticles confirmed that their T₂ MRI contrasts are also related to their varying polaron status.

C8: On the biosafety of the nanoparticles: the blood panel should also be performed at the 12 hours post injection; The authors should show the bio distribution and clearance of the particles after introduction.

Response R1-8: Thank you for making this point. As you suggested, evaluation of the biodistribution and clearance is important for biosafety investigations. However, the concept behind the current research is about the feasibility of using conjugated polymers in T₂ MRI. As mentioned previously, there is still much room for further improvement in the imaging abilities of conjugated polymers. A detailed evaluation is more necessary for contrast agents with more outstanding imaging abilities. Also the conditions for conducting animal experiments are not suitable for us at present. However, we would like to conduct detailed biodistribution investigations in our next research project.

C9: Fig. 2d: no circles are drawn to show the liver.

Response R1-9: Thank you for pointing out this error. We used a dotted circle to show the darkened region in the liver in Fig. 4a.

C10: Page 8: line 3 from the end of the paragraph: the normal range (Fig. 2a) should be Fig. 2b; the last sentence, (Fig. 2b and Table S1) should be Fig. 2c and Table S1.

C11: Fig.s4b misses the plot for partially complexed group.

Response R1-10 and R1-11: Thank you for pointing out these errors. We have revised the manuscript accordingly.

C12: Fig.s6: it seems there are two types of nanoparticles in the samples: one shows dark contrast and the others show light contrast. Do the authors know which is the

polymer nanoparticles? Could the other one be the inorganic nanoparticles with the iron metal?

Response R1-12: Thank you for your interesting questions. We believe that the light contrast can be attributed to residual surfactants. Although it is possible that some iron elements exist in our nanoparticles, they are unlikely to form inorganic particles as no oxidative conditions occur during the whole synthesis and washing process.

Reviewer #2:

General comment: In this manuscript, the authors claimed to develop a series of T₂ - weighted MRI active conjugated polymers (catechol-PPy, catechol-PEDOT or catechol-PANI nanoparticles) by using catechol derivatives to regulate the relative ratio of bipolarons to polarons. Although the concept is quite interesting, data presented here are not sufficient to fully support the proposed mechanism. And the proposed mechanism is ambiguous that lacks detailed explanation. Also, the imaging properties of these nanoparticles are not properly validated. Thus, a major revision is needed before its publication.

Response R2: Many thanks for reviewing our paper. In the revised version, we have emphasized the concept of utilizing conjugated polymers for T₂ MRI. The likely origin of T₂ MRI contrast in PPy nanoparticles is now explained in detail. We hope the revised manuscript now addresses your concerns.

C1: The authors should provide the specific chemical structures or at least the proposed chemical structures of catechol-PPy, catechol-PEDOT and catechol-PANI. Also, corresponding chemical characterization (e.g. ¹H NMR and gel permeation chromatography) should be performed to validate the synthesis of these compounds. In particular, please provide data to confirm the existence and the structure of the so called “oxidized species” in the catechol-polymer complex.

Response R2-1: Thank you for pointing out this important issue. The chemical

structures of oxidized forms of PPy, PEDOT and PANI have been added to the revised manuscript (Fig. 1 and Fig. S5). The existence of oxidized species was confirmed using UV-Vis-NIR absorption spectroscopy in which the polaron band (range 700-900 nm) and the bipolaron band (very broad, ranges above 1,000 nm) were detected for catechol-PPy/PEDOT/PANI nanoparticles. Moreover, the catechol-polymers did not form in our nanoparticles because the characteristic peaks of C=C stretching benzoic group around $1,625\text{ cm}^{-1}$ was not observed in the FTIR spectra of both pristine PPy and catechol-PPy nanoparticles.

Corresponding content in the revised manuscript:

Page 6, paragraph 1:

Response R2-1: Next, the impact of incorporated catechol derivatives should be excluded because they have been reported to generate T_2 MRI contrast using a high-field (9.4 T) MRI system¹⁷. Therefore, we analyzed the content of catechol derivatives in our catechol-PPy nanoparticles, as well as the T_2 relaxation times of catechol derivatives using a low-field (0.5 T) MRI system. First, the amounts of catechol derivatives in as-synthesized catechol-PPy nanoparticles were extremely low according to their FTIR spectra (Fig. S2a). This is because the FTIR spectra of pristine PPy and catechol-PPy nanoparticles are identical and the characteristic peak of the C=C stretching bond of the benzoic group around $1,625\text{ cm}^{-1}$ was not observed in any of the catechol-PPy nanoparticles. Also, catechol derivatives produce little shortening effect under a low-field (0.5 T) MRI system used in our research (Figure S2b). After excluding the possibility of iron elements and catechol derivatives for generating T_2 MRI contrast, we speculated that the T_2 MRI contrast can only be generated by paramagnetic oxidized polypyrrole instead of the catechol-polypyrrole complex. Such oxidized species, named polarons, are paramagnetic because they possess one unpaired electron.

C2: Catechols as a type of phenols have been reported as a family of T_2 -weighed MRI contrast agents (e.g. Chemistry, 2018, 24, 1259-1263). In this manuscript, it is

confusing and difficult to confirm in the catechol-polymer complex that whether it is the polymer or the catechol itself to play the major role as the MRI contrast agent. The authors should provide T_2 -weighted MRI signals from these catechols as control.

Response R2-2: This is an important factor in analyzing the origin of T_2 MRI contrast in our nanoparticles. First, we excluded the presence of catechol derivatives in the PPy nanoparticles by FTIR analysis. Also, the T_2 relaxation times of catechol derivatives was negligible in the low field MRI system used in our research. These T_2 relaxation times have been added as control groups in the revised manuscript.

Corresponding content in the revised manuscript:

Page 6, paragraph 1:

Response R2-2: Next, the impact of incorporated catechol derivatives was excluded because they have been reported to generate T_2 MRI contrast using a high-field (9.4 T) MRI system¹⁷. Therefore, we analyzed the content of catechol derivatives in our catechol-PPy nanoparticles, as well as the T_2 relaxation times of catechol derivatives using a low-field (0.5 T) MRI system. First, the amounts of catechol derivatives in as-synthesized catechol-PPy nanoparticles was extremely low according to their FTIR spectra (Fig. S2a). This is because the FTIR spectra of pristine PPy and catechol-PPy nanoparticles are identical and the characteristic peak of the C=C stretching bond of the benzoic group around $1,625\text{cm}^{-1}$ was not observed in any of the catechol-PPy nanoparticles. Also, catechol derivatives produce little shortening effect under the low-field (0.5 T) MRI system used in our research (Figure S2b). After excluding the possibility of iron elements and catechol derivatives for generating T_2 MRI contrast, we speculated that T_2 MRI contrast could only be generated by paramagnetic oxidized polypyrrole instead of the catechol-polypyrrole complex. Such oxidized species, named polarons, are paramagnetic because they possess one unpaired electron.

C3: Given the central topic of this work is about metal-free MRI contrast agents, the

authors should mention in the introduction the description about phenols (catechols included) for chemical exchange saturation transfer (CEST) MRI.

Response R2-3: Thank you for your valuable suggestions. We have accordingly rewritten the Introduction in the revised manuscript. The non-metallic agents are introduced in detail, including phenols for CEST MRI. Besides, we classified phenols as nonmetallic agents for clinically applicable T₂ MRI because phenols can shorten the T₂ relaxation times by CEST process according to the article mentioned in R2-C2.

Corresponding content in the revised manuscript:

Page 3, paragraph 1

Response R2-3: Until now, the only organic agents that can be used in clinically used MRI techniques are phenols and nitroxide radical molecules^{16,17}.

C4: T₂ -weighted MRI signals and images from catechol-PPy, catechol-PEDOT and catechol-PANI nanoparticles should be provided in the manuscript. Also, NIR-II PA signals and images of these nanoparticles should be provided.

Response R2-4: Thank you for pointing this issue out. T₂ MRI signals of catechol-PPy/PANI/PEDOT nanoparticles were measured as their T₂ relaxation times at the same mass concentrations. The T₂ relaxation times of PPy nanoparticles have been added to Table 1 and those of PEDOT/PANI nanoparticles to Fig. 3e in the revised manuscript. In addition, the NIR-PA images of PPy nanoparticles has been added to Fig. 1d. To emphasize the T₂ MRI contrast of the conjugated polymers, we reduced the discussion about NIR-PA. In consideration of the positive connections between the absorption bipolaron band and PA signals in the NIR-II region, the discussion of PA signals in PEDOT and PANI nanoparticles has been omitted from the revised manuscript.

C5: The relative T₂ MRI signal intensity of catechol-PPy, catechol-PEDOT or catechol-PANI nanoparticles should be benchmarked with the broadly reported contrast agents such as superparamagnetic iron-oxide NPs (SPIO-NPs) to show their potential in MRI

imaging.

Response R2-5: Thank you for your very helpful suggestion. Accordingly, we used the broadly reported T₂ MRI agents, superparamagnetic Fe₃O₄ nanoparticles, as a benchmark for the MRI properties of our conjugated polymer nanoparticles. The r₂ relaxivity of Fe₃O₄ nanoparticles is about 9.9 mL/mg·s which, as expected, was higher than for our conjugated polymer agents. This is because the fraction of paramagnetic species in conjugated polymers (about 10⁻⁹ mmol/mg) is lower than that in Fe₃O₄ nanoparticles (about 10⁻⁵ mmol/mg). The concept behind this research was the introduction of a new class of T₂ MRI agents rather than enhancing the imaging abilities of existing MRI agents. Further improving the MRI properties of conjugated polymers can certainly be achieved during future research by tuning their molecular structures.

Corresponding content in the revised manuscript:

Page 15, paragraph 2:

Response R2-5: Furthermore, this research can be a reference for the development of a series of nonmetallic T₂ MRI agents, which are not limited by particular molecular structures. This is because the oxidized forms are a common phenomenon in conjugated polymers of various molecular structures. Therefore, the imaging abilities of conjugated polymers, although not comparable with other newly developed inorganic T₂ MRI agents (Fig. S8), can certainly be improved by further tuning the structures of conjugated polymers.

C6: Quantification and statistical analysis of Fig. 2d and 2e should be performed.

Response R2-6: Thanks for your valuable suggestions. The quantification and statistical analysis of T₂ MRI contrast and PA signals are now provided in Fig. 4b and 4d of the revised manuscript.

Reviewer #3:

General comments: In this work, the authors reported T₂-weighted MRI active polypyrrole nanoparticles and correlated their paramagnetism with the formation of bipolaron. Although using CPN as MRI contrast agent is interesting, however, the manuscript could hardly meet the high criteria for publication in nature communication. Below are my concerns and suggestions.

Response R3: Many thanks for reviewing our manuscript. We have thoroughly revised manuscript by emphasizing the concept that activating conjugated polymers in T₂ MRI. According to our knowledge, this concept is firstly proposed and realized in this research. We consider that the novelty of this manuscript is suitable for publication in Nature Communication. Also according to your important suggestions, we added experiments and explanations to elucidate the origin of the T₂ MRI contrast in our nanoparticles. Hope the revision can meet the criteria for publication in *Nature Communication*.

C1: The reason for the r₂ relaxivity of DHBN-PPy NPs being much smaller than the other catechol-PPy NPs should be clearly explained.

Response R3-1: Thanks for pointing out these issues. We have briefly reviewed the possible mechanisms involving catechol derivatives. The radical trapping ability of catechol derivatives plays a key role in regulating bipolaron formation. This ability is related to the interactions between the HOMO of catechol derivatives and polarons. For DA, CA and DHBA, their HOMO are distributed on the hydroxyl group and can easily interact with polarons. However, for DHBN with a strong electron withdrawing group, -CN, its HOMO is distributed on the entire molecule so that it can hardly trap a polaron. Therefore, a significant decrease of r₂ relaxivity was observed in DHBN-PPy NPs.

Corresponding content in the revised manuscript:

Page 10, paragraph 1

Response R3-1: For catechol derivatives with strong electron withdrawing groups (e.g. DHBN with a cyano group), the radical trapping ability of their phenol

groups is very weak (Arabian Journal of Chemistry, 2017, 10, S1703-S1710.) so that they only produce a weak regulating effect on bipolaron formation.

C2: Considering the *R*-value of DHBA is moderately higher than the other catechol-PPy NPs, the reason for its r_2 relaxivity of polaron to be much higher than the others should be explained.

Response R3-2: Thank you for raising this important question. We attribute such inconsistency to different measurement technologies for these two parameters. The *R*-value is a semi-quantified parameter that represents the variation trend of the relative content of bipolarons vs polarons. The r_2 relaxivity of polaron is calculated from the T_2 relaxation times and polaron content. Both data are quantified parameters. Therefore, the exact variation in the degree of *R*-values and r_2 relaxivities of polarons strictly cannot be correlated with any degree of confidence.

Corresponding content in the revised manuscript:

Page 8, paragraph 1:

Response R3-2: The *R*-value is used to represent the variation trend of the relative content of bipolarons vs polarons, rather than the actual ratio in PPy nanoparticles.

C3: The author also fabricated PEDOT and PANI NPs and analyze their bipolarons content. Why their T_2 relaxation time measurement result was not shown? Their MRI performance are important to support the bipolaron theory.

Response R3-3: Thank you for your valuable suggestions. The MRI performances of PEDOT and PANI nanoparticles have been added to the revised manuscript. Although the shortening effects of T_2 relaxation times in catechols-PEDOT/PANI are not as significant as those in PPy, the variation of T_2 relaxation times vs bipolaron content provides evidence for our “bipolaron theory”.

Corresponding content in the revised manuscript:

Page 12, paragraph 2:

Response R3-3: Next, we investigated the correlation between increased bipolaron content and T_2 MRI contrast in PEDOT and PANI nanoparticles. As expected, the T_2 relaxation times were still shortened in the catechol-PANI/PEDOT nanoparticles with increased bipolaron content (Fig. 3e). Those results revealed that bipolaron content is a key parameters for T_2 MRI contrast. It can either activate or enhance the T_2 MRI contrast of conjugated polymers. Moreover, the different T_2 relaxation times of PANI and PEDOT nanoparticles revealed that the MRI properties of conjugated polymers are also impacted by molecular structures. Therefore, the T_2 MRI contrast have great potential for further improvement by elaborating their molecular structures.

C4: The UV-Vis-NIR absorption spectra of pristine PPy should be provided and taken into the discussion of bipolarons generation.

Response R3-4: Many thanks for pointing out this issue. Since the colloidal stability of pristine PPy nanoparticles in aqueous solutions cannot meet the requirements for measuring their absorption spectrum, the absorption spectra of different PPy nanoparticles cannot be measured under the same conditions. This is the reason why we introduced *R*-values to represent the relative content of bipolarons vs polarons.

C5: It is not clear why PVA-PPy shows much lower amount of bipolarons as compared with catechol-PPys.

Response R3-5: Thank you for raising this important question. The bipolaron amount in PVA-PPy is low because the very broad bipolaron band above 1,000 nm is not obvious in the UV-Vis-NIR absorption spectrum. For PPy with increased bipolaron content, the relative absorbance above 1,000 nm (bipolaron band) will be higher than that around 800 nm (polaron band). We have updated our explanations about the interactions between catechol and polarons/bipolarons.

The data for PVA-PPy are not necessary and have been omitted from the revised manuscript.

C6: The author should do a comparison of MRI imaging between PPy NPs and commercial MRI contrast.

Response R3-6: Thank you for your valuable suggestion. We used FDA-proven, superparamagnetic Fe₃O₄ nanoparticles synthesized by the hydrothermal method as a standard reference for the MRI properties of our conjugated polymer nanoparticles. The r₂ relaxivity of Fe₃O₄ nanoparticles is about 9.9 mL/mg·s which as expected, was higher than for the conjugated polymer agents. This is because the fraction of paramagnetic species in conjugated polymers (about 10⁻⁹ mol/mg) is lower than that in Fe₃O₄ nanoparticles (about 10⁻⁵ mol/mg). The concept behind the present research was the introduction of a new class of T₂ MRI agents rather than enhancing the imaging abilities of existing MRI agents. Further improvements in the MRI properties of conjugated polymers can certainly be achieved in future research by tuning their molecular structures.

Corresponding content in the revised manuscript:

Page 15, paragraph 2:

Response R3-6: Furthermore, this research can be a reference for the development of a series of nonmetallic T₂ MRI agents, which are not limited by particular molecular structures. This is because the oxidized forms are common phenomena in conjugated polymers with various molecular structures. Therefore, the imaging abilities of conjugated polymers, although not comparable with other newly developed inorganic T₂ MRI agents (Fig. S8), can certainly be improved by further tuning the structures of conjugated polymers.

C7: The analysis of substitutional group effect on the formation of bipolar is very confusing. When the substitutional groups of catechol derivatives varied from electron donating to neutral and electron withdrawing, the radical scavenger effect decreased

and thus polymerization process was less affected, leading to more polarons. The authors should clearly explain the moderate radical scavenger effect.

Response R3-7: Thank you for your helpful suggestions. We have rewritten this section in the revised manuscript. The interactions between catechol derivatives and polarons are divided into two types: the first one is the radical scavenging effect, which can directly terminate polarons; the second is the radical trapping effect, which can temporally “trap” the polarons. Please see the detailed explanation in the revised manuscript, which is provided below:

Corresponding content in the revised manuscript:

Page 9, paragraph 1:

Response R3-7: Then, the bipolaron formation is regulated by the radical trapping ability of phenol groups on the catechol derivatives, an ability that is changed by substituted groups³³. Therefore, catechol-PPy nanoparticles with different substituted groups exhibited different *R*-values. When substituted with electron donating moieties (e.g. dopamine), the catechol derivatives tend to exhibit radical scavenger effects that can directly terminate polarons. When substituted with electron withdrawing moieties (e.g. DHBA), the catechol derivatives tend to trap instead of terminate polarons³³ so that catechol derivatives can regulate the bipolaron formation process. We speculate that the radical trapping ability functions by increasing the probability of polarons forming bipolarons. For catechol derivatives with strong electron withdrawing groups (e.g. DHBN with a cyano group), the radical trapping ability of their phenol groups is very weak³³ so that they only produce a weak regulating effect on bipolaron formation. Therefore, the maximum *R*-value lies in the DHBA group. Also, it should be noted that because both radical scavenging and trapping processes exist for catechol derivatives, the *R*-values strictly can hardly vary from two polarons into one bipolaron. Hence, the variation of *R*-values appears to be moderate in comparison with the polaron content (Table 1). Although the detailed polymerization process remains to be unequivocally established, the relationship between substitutional

groups and *R*-values has proven that the radical trapping ability is the key factor that regulates bipolaron formation.

C8: Figure 1a is very confusing. The structures of synthesized catechol-PPy should be given and the interaction between catechol and polaron/bipolaron should be highlighted.

Response R3-8: Thank you for your valuable comments. The chemical structures of polarons and bipolarons has been added to Fig. 1 in the revised manuscript. Also, we decided to focus on the T₂ MRI contrast of conjugated polymers rather than the detailed interactions between catechol derivatives and polypyrrole. Hence a simplified mechanism has been included in the revised Fig. 1. Also please refer to our explanations about such interactions in the “corresponding content in the revised manuscript” for R7.

C9: Figure 1b, the quality of TEM image is low. And the average size is obviously not 20 nm as described but 10-15 nm.

Response R3-9: Many thanks for pointing out this issue. The electron contrast of polymer nanoparticles in TEM is reduced because of their low electron density. We used 20 nm in the previous version by including the ambiguous boundary around the nanoparticles. According to your valuable suggestions, the average size of nanoparticles has been modified to 15 nm using the clear boundary of the nanoparticles.

C10: The calculation method for polaron content should be provided.

Response R3-10: Thank you for raising this issue. The polaron amount is measured using the Spin counting tool of Xenon software, after a factory calibration of the cavity response function. The uncertainty of the spin concentration was estimated to be within ~10%

Corresponding content in the manuscript:

Page 7, caption for Table 1:

Response R3-10: The spin counting tool of the Xenon software was used to determine the spin density (polaron amount) of PPy samples after a factory calibration of the cavity response function. The uncertainty of the spin concentration was estimated to be within ~10%

C11: Many related literature should be cited, especially in the introduction part.

Response R3-11: Thank you very much for this suggestion. We have rewritten the introduction accordingly. The importance of developing nonmetallic MRI agents and current nonmetallic agents are discussed in the revised manuscript.

Response R1-R2-R3:

Once more, many thanks to all the reviewers for all your kind questions and most helpful comments that have, without doubt, greatly improved the quality and clarity of our revised manuscript.

REVIEWER COMMENTS

Reviewer #1 (Remarks to the Author):

The revised manuscript is still confusing while some of my prior concerns are clarified.

The first is the provided explanation for the observed magnetism. Figure 1 shows bipolarons are responsible for the r_2 MRI relaxivity, but it also indicates that polarons are paramagnetic. This is confusing and inconsistent. In the authors' reply, they revised the explanation and proposed that the polaron status and content, and their variation by dipolarons determine the T2 relaxivity. However, both terms are not carefully defined in the paper and I have a hard time following the argument. How many variations in their status exist? In fact, the definition of polarons and dipolaron is inconsistent. On Page 8 it explains bipolaron is formed by two polarons, but Fig. 1 shows their nearly identical structures except the charge number and free radical.

Next, the authors provided additional data to support their claim that the observed T2 relaxivity didn't come from the iron ions left in the nanoparticles from the synthesis. They showed that both pristine PPy nanoparticles and catechol-PPy nanoparticles contained 1-2% Fe but the former didn't show T2 shortening effect. However, the iron oxidation status in both nanoparticles may not be the same. Different iron oxide nanoparticles can show different relaxivity even the same amount of iron. Since the r_2 relaxivity is around a few percentage of Fe₃O₄ nanoparticles, the trapped Fe remains potential suspect for the observed T2 shortening effect.

The work just does seem to be suitable to the general readership of Nature Comm but a more specialized journal, while the main claim is still up to be proved.

Reviewer #2 (Remarks to the Author):

We appreciate the efforts that the authors made to improving the manuscript. Whereas the following issues are still remaining:

1. The changes of charge in Fig.1 for PPy after catechol treatments should be characterized and proved.
2. ¹H NMR spectra for catechol-PPy, catechol-PEDOT and catechol-PANI in comparison with PPy, PEDOT and PANI should be provided as requested. The loss of protons after catechol treatments should be clearly indicated from the spectra.
3. In Fig.2c, besides PPy, the Raman spectra of DHBN-PPy, DHBA-PPy, CA-PPy, DA-PPy should be provided in comparison with PPy. Likewise, spectra for all these groups should be provided in Fig. 2d and e.
4. TEM images for catechol-PEDOT and catechol-PANI nanoparticles should be replaced with new images with higher resolution and less impurities in the background.
5. For some key experiments (e.g. Fig. 3e, Fig. 4b, Fig. 4d, Fig. S2b, Fig. S4b), the data should be measured in triplicate, the plot should present all the data measured together with mean \pm standard deviation, and statistical analysis should be performed between the experimental and control groups.
6. In stability study (Fig. S6), the authors did not show the difference/comparison of hydrodynamic size and oxidized species before and after storage of DHBA-PPy@DSPE-PEG. Also, the term for such storage was not clearly indicated. The authors should perform the stability study lasting for at least a week.
7. The comparison of in vivo imaging performance between the conjugated polymer nanoparticles in this work and Fe₃O₄ nanoparticles should be provided in the manuscript rather than supporting information.
8. Typo: Fig. 4f

Comments from reviewer #2 on the response to reviewer #3;

The reviewer appreciates the efforts made by the authors to revise the manuscript. Most questions were answered in reason. Nevertheless, the following issues persist:

1. For C4, the authors replied that "the colloidal stability of pristine PPy nanoparticles in aqueous solutions cannot meet the requirements for measuring their absorption spectrum." It could be speculated that the pristine PPy nanoparticles suffer from poor water solubility. Considering that for in vivo imaging the authors prepared DHBA-PPy@DSPE-PEG nanoparticles, the authors might consider providing absorption spectra of pristine PPy nanoparticles and bipolaron nanoparticle series after formulation with this surfactant.

2. In the response to C6, the authors mentioned that the r_2 relaxivity of Fe₃O₄ nanoparticles is about 9.9 mL/mg·s, which was approximately 24 folds of that for DHBA-PPy (0.415 mL/mg·s). We agree with the authors' opinion that the current work is to introduce a new class of T2 MRI agents rather than reporting something to outperform commercialized MRI contrast agents. However, the new class of contrast agents hardly showed superiorities over existing MRI contrast agents in clinical use. Essentially speaking, these bipolaron molecules were designed to serve as contrast agents, wherein the imaging performance in practice cannot be ignored as standards of evaluation. In other words, the advantages of these bipolaron molecules for MRI imaging over existing contrast agents should be emphasized.

3. In C8, reviewer 3 asked for a revision of Figure 1a. As requested in this opinion, I think the authors should consider illustrating the interactions between catechol and polaron/bipolaron molecules in Figure 1a, in addition to providing the explanations in the manuscript.

Reviewer #1:

General comments (C1): The revised manuscript is still confusing while some of my prior concerns are clarified.

The work just does seem to be suitable to the general readership of Nature Comm but a more specialized journal, while the main claim is still up to be proved.

Response (R1): Thanks for reviewing our paper and your comments as well. In the revision, we have updated the explanation for the observed magnetism and provided additional experiments to characterize residual iron elements. We hope the revision can now address your concerns.

Comment 1-1 (C1-1): The first is the provided explanation for the observed magnetism. Figure 1 shows bipolarons are responsible for their r_2 MRI relaxivity, but it also indicates that polarons are paramagnetic. This is confusing and inconsistent. In the authors' reply, they revised the explanation and proposed that the polaron status and content, and their variation by dipolarons determine the T_2 relaxivity. However, both terms are not carefully defined in the paper and I have a hard time following the argument. How many variations in their status exist? In fact, the definition of polarons and dipolaron is inconsistent. On Page 8 it explains bipolaron is formed by two polarons, but Fig. 1 shows their nearly identical structures except the charge number and free radical.

Response R1-1: Many thanks for pointing above issues. Firstly, the definitions of polarons and bipolarons have been added in the revision. Polarons are generated when oxidative agents remove one electron from polymer's valence band and create a polymer radical cation. When polaron content reaches high level, the radical cations tend to combine to form bipolarons. The schematic in Fig. 1 has been revised.

Corresponding content in the revision:

Page 7, paragraph 1

“The paramagnetic species in PPy and other conjugated polymers are polymer radical cations called polarons. Polarons are formed when oxidative agents remove an electron from the valence band of conjugated polymers.”

Page 9, paragraph 1

“Bipolarons are polymer dications formed by two coupled polarons, when polaron concentration increase.²⁷”

Next, the explanation about observed magnetism has been revised. Generally, T₂ MRI contrast of agents is impacted by two factors: the content of paramagnetic species and their status. In conjugated polymers, paramagnetic species are polarons. Their status are quantified using r₂ relaxivity of polarons. In this research, we found that the variation of polaron status is the key factor for utilizing conjugated polymers for T₂ MRI. Such variation of status can be divided into two steps: (1) The interactions between unpaired electrons and water protons are the basis of contrast enhanced T₂ MRI. However, in pristine conjugated polymers, the unpaired electrons on polarons are in strong “spin-spin” interactions with other polarons. Therefore, these unpaired electrons cannot interact with water protons so that pristine conjugated polymers only exhibit negligible T₂ MRI ability. In this research, our method can “release” polarons from above strong “spin-spin” interactions using radical trapping agents, catechol derivatives. (2) When the concentration of polarons is high, they tend to form bipolarons around polarons. Our method increases bipolaron content around polarons. The electrostatic repulsion brought by bipolarons can further elevated the interactions between unpaired electrons on polarons and water protons. While this research is about utilizing conjugated polymers for T₂ MRI, quantifying the exact extent of above interactions is not the main purpose of this research and we would like to investigate this issue in our next research.

Corresponding content in the revision:

Page 9, paragraph 1

*“To further get insight into how polaron status varied, we used DA-PPy nanoparticles to investigate the structural parameters that may be changed in catechol-PPy nanoparticles prepared with our “partially complexed” method. Different DA-PPy nanoparticles were prepared with different fractions of ferric ions complexed with DA (DA/Fe³⁺). With DA/Fe³⁺ increases, the polaron content also increases because the ESR signal intensity of these nanoparticles elevated (**Fig. 2c**). However, only 0.1/0.3/0.6DA-PPy nanoparticles exhibit significant shortening effect on the T₂ relaxation times (**Fig. 2d**). Such inconsistency indicates the variation of polaron status are further related to other structural parameters. Besides the variation of the ESR signal intensity, the absorption spectra of different DA-PPy nanoparticles also change obviously in the near-infrared region (**Fig. 2e**). The relative absorbance in the second near-infrared (NIR-II) region (> 1000 nm) versus that in the NIR-I region (700 nm~1000 nm) of 0.1/0.3/0.6DA-PPy nanoparticles is higher than that of 1.2/1.8/2.3DA-PPy nanoparticles. For PPy, the NIR-I absorbance is assigned to polarons and NIR-II absorbance are assigned to bipolarons. Bipolarons are polymer dications formed by two coupled polarons, when polaron concentration increase.²⁷ In spite of the absorption spectra of PPy, we also used XPS analysis to verify the presence of bipolarons by deconvoluting the N 1s peak of all five kinds of PPy nanoparticles. The N 1s spectra can be deconvoluted into four Gaussian peaks centered at 398.4, 399.8, 400.4, and 402.8 eV (**Fig. S3**). The peaks at 398.4 and 399.8 eV are ascribed to neutral –N= and –NH– motifs in the backbone of PPy chains, while the high-binding-energy tails centered at 400.4 and 402.8 eV are ascribed to charged –N⁺H– and –N⁺= motifs. The total protonation levels are contributed by both polarons and bipolarons. For pristine and catechol-PPy nanoparticles, their protonation levels varies in a different trend compared with the polaron contents of them (**Table 1** and **Table S1**). Thus we inferred that the bipolaron content also changed and contributed to the total protonation levels. Since bipolarons exhibit two positive charges, the electrostatic repulsion interactions between polarons (cation) and bipolarons (dication) might change the polaron status.”*

“These observations confirm that variation in the relative content of bipolarons changed the status of the paramagnetic polarons. The increased relative bipolaron content enhance the electrostatic repulsion around polarons. Thus the interactions between polarons are further reduced and interactions between polarons and water protons are further elevated. According to above studies, we conclude that the T_2 MRI contrasts of catechol-PPy nanoparticles are associated with two steps: (1) Polarons should be released from the strong “spin-spin” interactions with other polarons. In this research, catechol derivatives can fulfill this requirement by their radical trapping ability. (2) Bipolarons are required to be formed around polarons to further enhance the interactions between polarons and water protons. The mechanism of the “partially complexed” method regulating bipolaron content remains to be investigated.”

C1-2: Next, the authors provided additional data to support their claim that the observed T_2 relaxivity didn't come from the iron ions left in the nanoparticles from the synthesis. They showed that both pristine PPy nanoparticles and catechol-PPy nanoparticles contained 1-2% Fe but the former didn't show T_2 shortening effect. However, the iron oxidation status in both nanoparticles may not be the same. Different iron oxide nanoparticles can show different relaxivity even the same amount of iron. Since the r_2 relaxivity is around a few percentage of Fe_3O_4 nanoparticles, the trapped Fe remains potential suspect for the observed T_2 shortening effect.

Response R1-2: Thanks for raising this important question. The contribution of trapped Fe to the T_2 MRI properties of catechol-PPy cannot be entirely excluded as you mentioned. However it may not be the determining factor for conjugated polymers generating T_2 MRI contrast. Firstly, we provided additional XPS results in the revision to support that residual Fe elements remain similar for both pristine and catechol-PPy nanoparticles. Then we have tried to characterize iron oxidation status. However, the iron oxidation status can hardly characterized by XPS because of the extremely low intensity of Fe elements. Thus we tried to use other experiments to support that residual iron elements is not the determining factor for activating conjugated polymers

in T₂ MRI. The iron oxidation status for DA-PPy nanoparticles prepared with different amount of dopamine nearly remain the same because dopamine cannot reduce ferric ions. While T₂ MRI signals of these DA-PPy nanoparticles exhibited obvious variations (**Fig. 2d**). Moreover the paramagnetism of conjugated polymers have been proved to be a polaron related properties, we conclude that the different T₂ MRI properties between pristine and catechol PPy nanoparticles are caused by the variation of polarons.

Corresponding content in the revision:

Page 6, paragraph 1

*“Such low levels of iron elements in pristine and catechol-PPy nanoparticles are also observed in their XPS spectra (**Fig. S1b**).”*

Reviewer #2:

General Comments C1: We appreciate the efforts that the authors made to improving the manuscript. Whereas the following issues are still remaining:

Response R2: Thanks for reviewing our paper. We have modified the manuscript according to following comments. We hope the revision can address following issues.

C2-1: The changes of charge in Fig.1 for PPy after catechol treatments should be characterized and proved.

R2-1: Many thanks for pointing out this issue. We used XPS to characterize the changes of charge in PPy by curving N 1s peak. The peaks with lower binding energies represent neutral $-N=-/NH=$ motifs and those with higher binding energies represent charged $-N^+=-/NH^+=$ motifs. The fraction of charged motifs in catechol-PPy nanoparticles varied differently with polaron content. This results not proved that catechol treatments change the fraction of charged motifs but confirmed the presence of bipolarons. This is because both polarons and bipolarons contain charged motifs.

Corresponding content in the revision:

Page 10, paragraph 1

"In spite of the absorption spectra of PPy, we also used XPS analysis to verify the presence of bipolarons by deconvoluting the N 1s peak of all five kinds of PPy nanoparticles. The N 1s spectra can be deconvoluted into four Gaussian peaks centered at 398.4, 399.8, 400.4, and 402.8 eV (Fig. S3). The peaks at 398.4 and 399.8 eV are ascribed to neutral $-N=$ and $-NH-$ motifs in the backbone of PPy chains, while the high-binding-energy tails centered at 400.4 and 402.8 eV are ascribed to charged $-N^+H-$ and $-N^+=$ motifs. The total protonation levels are contributed by both polarons and bipolarons. For pristine and catechol-PPy nanoparticles, their protonation levels varies in a different trend compared with the polaron contents of them (Table 1 and Table

S1). Thus we inferred that the bipolaron content also changed and contributed to the total protonation levels.”

C2-2: ^1H NMR spectra for catechol-PPy, catechol-PEDOT and catechol-PANI in comparison with PPy, PEDOT and PANI should be provided as requested. The loss of protons after catechol treatments should be clearly indicated from the spectra.

R2-2: Thanks for your suggestion. Since these three polymers are insoluble in all common solvents, we used solid-state ^1H NMR to characterize all six kinds of nanoparticles as you mentioned. However, the paramagnetism of these polymers can cause inhomogeneity of the magnetic field. Such effect can significantly broaden the chemical shifts so that it is hard to extinguish chemical shifts of different H atoms. Therefore, we decided to exclude these data in the revision. Please find the NMR spectra of these polymers below.

C2-3: In Fig.2c, besides PPy, the Raman spectra of DHBN-PPy, DHBA-PPy, CA-PPy, DA-PPy should be provided in comparison with PPy. Likewise, spectra for all these groups should be provided in Fig. 2d and e.

R2-3: Thanks for the suggestion. We have combined Raman spectra of pristine and catechol-PPy nanoparticles into Fig. 2g

C2-4: TEM images for catechol-PEDOT and catechol-PANI nanoparticles should be replaced with new images with higher resolution and less impurities in the background.

R2-4: Thanks for raising this issue. TEM images of these two nanoparticles have been updated in Fig **3a** and **3c**. The resolution and background are optimized. While these nanoparticles are all organic ones, which usually have limited contrast and resolution in comparison with inorganic nanomaterials.

C2-5: For some key experiments (e.g. Fig. 3e, Fig. 4b, Fig. 4d, Fig. S2b, Fig. S4b), the data should be measured in triplicate, the plot should present all the data measured together with mean \pm standard deviation, and statistical analysis should be performed between the experimental and control groups.

R2-5: Many thanks for pointing this issue, all these experiments and figures have been revised accordingly. For **Fig. S4b** in the previous version, we replaced this picture with T_2 relaxation times of partially complexed group and complexed group in **Fig. 2d** with 0.1DA-PPy and 2.3DA-PPy.

C2-6: In stability study (Fig. S6), the authors did not show the difference/comparison of hydrodynamic size and oxidized species before and after storage of DHBA-PPy@DSPE-PEG. Also, the term for such storage was not clearly indicated. The authors should perform the stability study lasting for at least a week.

R2-6: Thanks for raising this question. We performed the stability test at room temperature under sterilized environment for one week. The colloidal stability remains with similar hydrodynamic sizes and absorption spectra in **Fig. S6**.

Corresponding content in the revision:

Page 15, paragraph 2

“After incubating DHBA-PPy@DSPE-PEG nanoparticles in complete DMEM culture medium for 7 days under sterilized environment at room temperature, the absorption spectrum as well as the hydrodynamic size of the nanoparticles remained similar compared with those in phosphate buffer saline”

C2-7: The comparison of in vivo imaging performance between the conjugated polymer nanoparticles in this work and Fe₃O₄ nanoparticles should be provided in the manuscript rather than supporting information.

R2-7: This is an important suggestion for this work. We have performed in vivo imaging using DHBA-PPy and Fe₃O₄ nanoparticles in **Fig. 4a** and **4b**. As expected, the in vivo performance of commercially available T₂ agents is superior to our organic T₂ MRI agents because unpaired electrons in Fe₃O₄ nanoparticles are 4-5 orders higher than those in DHBA-PPy nanoparticles. While this paper aims to investigate the feasibility of conjugated polymers for MRI rather than developing organic agents with Fe₃O₄ comparable MRI abilities. For further enhancing the MRI abilities of conjugated polymers, we considered how to increase polaron content and retain their “MRI-active” status should be evaluated.

Corresponding content in the revision:

Page 17, paragraph 1

*“To evaluate the applications of our conjugated polymer based T₂ MRI agents, we used Fe₃O₄ nanoparticles as a standard reference for in vivo imaging. As expected, the T₂ MRI contrast as well as the r₂ relaxivity of conjugated polymers are fairly weak in comparison with those of Fe₃O₄ nanoparticles (**Fig. 4b** and **Fig. S8**). This is because the concentration of unpaired electrons in Fe₃O₄ nanoparticles is about 4-5 orders higher than that of PPy nanoparticles. Hence, finding new method to increase the polaron content and keep the “MRI-active” status of polarons unchanged is necessary for developing conjugated polymers based T₂ MRI agents with Fe₃O₄ comparable imaging contrasts.”*

C2-8: Typo: Fig. 4f

R2-8: Thanks for pointing this typo, we have revised that accordingly.

Comments from reviewer #2 on the response to reviewer #3;

C3: The reviewer appreciates the efforts made by the authors to revise the manuscript.

Most questions were answered in reason. Nevertheless, the following issues persist:

R3: Thanks again for reviewing this manuscript. We have modified the manuscript according to following comments. We hope the revision can address following issues.

C3-1: For C4, the authors replied that “the colloidal stability of pristine PPy nanoparticles in aqueous solutions cannot meet the requirements for measuring their absorption spectrum.” It could be speculated that the pristine PPy nanoparticles suffer from poor water solubility. Considering that for in vivo imaging the authors prepared DHBA-PPy@DSPE-PEG nanoparticles, the authors might consider providing absorption spectra of pristine PPy nanoparticles and bipolaron nanoparticle series after formulation with this surfactant.

R3-1: Many thanks for this suggestion. According to that, UV spectrum of pristine PPy@DSPE-PEG nanoparticles has been added into **Fig. S4**.

C3-2: In the response to C6, the authors mentioned that the r2 reflexivity of Fe₃O₄ nanoparticles is about 9.9 mL/mg·s, which was approximately 24 folds of that for DHBA-PPy (0.415 mL/mg·s). We agree with the authors’ opinion that the current work is to introduce a new class of T₂ MRI agents rather than reporting something to outperform commercialized MRI contrast agents. However, the new class of contrast agents hardly showed superiorities over existing MRI contrast agents in clinical use. Essentially speaking, these bipolaron molecules were designed to serve as contrast agents, wherein the imaging performance in practice cannot be ignored as standards of evaluation. In other words, the advantages of these bipolaron molecules for MRI imaging over existing contrast agents should be emphasized.

R3-2: Many thanks for this valuable suggestion. We have reconsidered the advantages of conjugated polymer based T₂ MRI agents. Til now, only nitroxide radical agents have been reported for in vivo T₂ MRI. However, these agents suffer from poor in vivo

stability. This is because the unpaired electrons of nitroxide radical agents are stabilized by steric hinderance and can be rapidly terminated by in vivo reductive pathways. Currently, only very few kinds of nitroxide radical based MRI agents can retain “MRI-active” state in vivo for hours. These agents are obtained by very complicated synthetic strategies because steric groups around unpaired electrons are strictly confined. However, stabilizing unpaired electrons can be easily achieved by conjugated polymers because they can delocalize unpaired electrons along the unique conjugated backbone. Utilizing the optical properties of polarons and bipolarons for in vivo imaging has been reported. Thus we propose that developing conjugated polymer based MRI agents can be a solution for the in vivo stability problem for organic MRI agents.

Corresponding content in the revision:

Page 3, paragraph 2:

“The MRI properties of nitroxide radical molecules come from their unpaired radical electrons, which are stabilized by steric hindrance. However, nitroxide radicals suffer from poor in vivo stability. This is because the paramagnetic radicals can be rapidly deactivated by in vivo radical scavenging pathways⁹. Current approaches to increase the in vivo stability of nitroxide radicals are elaborating the chemical structures by tedious synthetic processes. Thus very few nitroxide radicals based MRI agents exhibit “MRI-active” state for hours. Therefore, further developing organic MRI agents still requires finding new strategy to stabilize paramagnetic unpaired electrons.”

C3-3: In C8, reviewer 3 asked for a revision of Figure 1a. As requested in this opinion, I think the authors should consider illustrating the interactions between catechol and polaron/bipolaron molecules in Figure 1a, in addition to providing the explanations in the manuscript.

R3-3: Thanks for raising this issue. We have revised **Fig.1** by emphasizing the process of variation of polaron status.

Response R1-R2:

Once more, many thanks to all the reviewers for all your kind questions and most helpful comments that have, without doubt, greatly improved the quality and clarity of our revised manuscript.

REVIEWER COMMENTS

Reviewer #1 (Remarks to the Author):

The authors did another round of revision to improve the clarity of the work in addressing my previous comments. Just a couple minor suggestions on its presentation:

1. The revised Figure 1 is cited at the end of the introduction section but without much explanation. The caption states "partially complexed" method, but no information in the associated text. This figure presents the essential idea of the paper, and the authors may consider either move it to the end in the discussion or add more context when it is first introduced.
2. A few places cite Fig. 1f and 1g. They should be Fig. 2f & 2g.

Reviewer #4 (Remarks to the Author):

The authors have made efforts to revise the manuscript. However, some issues still remain to be clarified.

Regarding response R2-1:

XPS data have now been added to support the authors' claims. However, the quality of the XPS results and, especially, the analysis and presentation makes any conclusion derived from those data ambiguous:

- The authors did subtract the secondary electron background prior to fitting. The reader cannot see the full picture any more as the authors already predefine the number of components required to deconvolute the experimental results. There is a clear contrast in the presentation between the low and high binding energy side of the N1s peak. While, on the low binding energy side, the spectra are truncated (see fig S3c and d), on the high binding energy side, intensities are shown on a larger energy range intended to motivate the need of additional chemically shifted components. However, at the high binding energy side, the intensity could well be just background signal rather than representing chemically shifted components thus making the presence of the latter in that binding energy range questionable.
- The scatter in the original data is quite enormous. There is no experimental evidence for the presence of up to 4 chemically shifted components used by the authors to deconvolute the spectra. Consequently, any conclusions drawn from those 4 components do not have any significance.
- For a reliable analysis which can be followed by the reader, the original spectra should to be presented showing the background signal within a sufficient energy range to the left as well as to the right side of the N1s signal. Furthermore, the original data should be used for fitting without subtracting the background in an arbitrary way (see, e.g., the seminal paper by Wertheim and Diczynski, "LEAST-SQUARES ANALYSIS OF PHOTOEMISSION DATA", Journal of Electron Spectroscopy and Related Phenomena, 37 (1985), 57-67)

R2-4: The TEM image quality is still not good. The authors could zoom in and provide an enlarged figure as an inset to see the morphology and size of the nanoparticles.

R2-6: The authors have shown the hydrodynamic size (Fig S6, DLS data) in PBS buffer and in culture medium. The size of the nanoparticles is expected to be larger after coating with DSPE-PEG. However, after comparing the typical sizes observed in TEM (though they are in the dried state), for nanoparticles coated with DSPE-PEG of Mw=5000, the size observed is substantially large in PBS (irrespective of size changing or not after storage in the medium). In the preparation,

it is indicated that the nanoparticles (1 mg) coated with 10 mg DSPE-PEG by sonication for 10 minutes. The latter as they are surfactants, will also form aggregated structures and can contribute to DLS size measurements.

As there were no indications of washing steps after sonication, how did they remove the contribution from DSPE-PEG structures that were not coating the nanoparticles. The authors need to clarify this issue.

As a suggestion, if possible the authors could use cryo-TEM to see the lipid coated structure and compare the size before and after coating.

Reviewer #1:

General comments (C1): The authors did another round of revision to improve the clarity of the work in addressing my previous comments. Just a couple minor suggestions on its presentation:

Response (R1): Many thanks for reviewing our paper and positive comments as well. We have modified the manuscript according to following comments. We hope the revision can address following issues.

Comment 1-1 (C1-1): The revised Figure 1 is cited at the end of the introduction section but without much explanation. The caption states "partially complexed" method, but no information in the associated text. This figure presents the essential idea of the paper, and the authors may consider either move it to the end in the discussion or add more context when it is first introduced.

Response 1-1 (R1-1): Thanks for the suggestion. We have rearranged the figures accordingly. Figure 1 in the previous revision has been moved to Figure 2 in the current revision.

C1-2: A few places cite Fig. 1f and 1g. They should be Fig. 2f & 2g.

R1-2: Thanks for pointing these typo, we have accordingly revised our manuscript.

Reviewer #4:

General Comments (C4): The authors have made efforts to revise the manuscript. However, some issues still remain to be clarified.

R4: Many thanks for reviewing our paper. According to your valuable suggestions, we have made further efforts to revise the XPS data and TEM images. We hope the revision can clarify following issues.

C4-1: Regarding response R2-1:

XPS data have now been added to support the authors' claims. However, the quality of the XPS results and, especially, the analysis and presentation makes any conclusion derived from those data ambiguous:

- The authors did subtract the secondary electron background prior to fitting. The reader cannot see the full picture any more as the authors already predefine the number of components required to deconvolute the experimental results. There is a clear contrast in the presentation between the low and high binding energy side of the N1s peak. While, on the low binding energy side, the spectra are truncated (see fig S3c and d), on the high binding energy side, intensities are shown on a larger energy range intended to motivate the need of additional chemically shifted components. However, at the high binding energy side, the intensity could well be just background signal rather than representing chemically shifted components thus making the presence of the latter in that binding energy range questionable.
- The scatter in the original data is quite enormous. There is no experimental evidence for the presence of up to 4 chemically shifted components used by the authors to deconvolute the spectra. Consequently, any conclusions drawn from those 4 components do not have any significance.
- For a reliable analysis which can be followed by the reader, the original spectra should be presented showing the background signal within a sufficient energy range to the left as well as to the right side of the N1s signal. Furthermore, the original data should be used for fitting without subtracting the background in an arbitrary way (see, e.g.,

the seminal paper by Wertheim and Diczko, "LEAST-SQUARES ANALYSIS OF PHOTOEMISSION DATA", Journal of Electron Spectroscopy and Related Phenomena, 37 (1985), 57-67)

R4-1: Many thanks for pointing this important issue. Firstly, according to several existing researches, we deconvoluted N1s peak into four peaks to estimate the fraction of neutral and charged species in PPy nanoparticles. These researches have been added into Ref.28-30. Then, we have re-processed the XPS data according to this seminal paper without subtracting the background as shown in Fig.S3 and Table. S1. After processing, the fractions of neutral and charged species (in Table. S1) in different PPy nanoparticles varied similarly compared those in previous revision. Hence, we can confirmed that the variation of charged species is attributed to both polarons and bipolarons.

Corresponding content in the revision:

Page 9, paragraph 1

"According to previous researches²⁸⁻³⁰, the N 1s spectra can be deconvoluted into four peaks centered at 398.4, 399.8, 400.4, and 402.8 eV (Fig. S3)."

Ref28-30:

28. Song, Y. et al. Pushing the cycling stability limit of polypyrrole for supercapacitors. *Advanced Functional Materials* 25, 4626-4632 (2015).

29. Song, Y., Xu, J.-L. & Liu, X.-X. Electrochemical anchoring of dual doping polypyrrole on graphene sheets partially exfoliated from graphite foil for high-performance supercapacitor electrode. *Journal of power sources* 249, 48-58 (2014).

30. Zhu, M. et al. Highly flexible, freestanding supercapacitor electrode with enhanced performance obtained by hybridizing polypyrrole chains with MXene. *Advanced Energy Materials* 6, 1600969 (2016).

Page 9, paragraph 1

"For pristine and catechol-PPy nanoparticles, their protonation levels varies in a different trend compared with the polaron contents of them (Table 1 and Table S1).

For example, the protonation level of pristine PPy is 21.6%, which is lower than that of DA-PPy and DHBN-PPy. While the polaron content of pristine PPy is about one order higher than DA-PPy and DHBN-PPy. The inconsistent variation between protonation levels and polaron content suggests that the bipolaron content also changed and contributed to the total protonation levels.”

Corresponding content in the supplementary information:

Page 3, paragraph 1:

“The deconvolution of N1s peak was processed according to previous report without subtracting the background (Wertheim, G. K., & Dicenzo, S. B. (1985). Least-squares analysis of photoemission data. *Journal of electron spectroscopy and related phenomena*, 37, 57-67).”

Fig.S3

Table S1

	Neutral motifs	Charged motifs
Pristine PPy	78.4%	21.6%
DA-PPy	73.9%	26.1%
CA-PPy	78.9%	21.1%
DHBA-PPy	82.4%	17.6%
DHBN-PPy	73.5%	26.5%

C4-2: R2-4: The TEM image quality is still not good. The authors could zoom in and provide an enlarged figure as an inset to see the morphology and size of the nanoparticles.

R4-2: Thanks for this valuable suggestion. The zoomed TEM images of PANI and PEDOT nanoparticles have been inserted in Fig. 3a and 3c. Usually the contrast of conjugated polymer nanoparticles in TEM images is shallow in comparison with inorganic nanoparticles. Organic nanoparticles only contain light elements which attenuated electrons more weakly than inorganic ones under TEM imaging. Hence, the quality of TEM images for conjugated polymer nanoparticles have limited space for improvement. We have tried our best to improve the quality of TEM images.

Corresponding content in the revision:

Fig.3a and 3c

C4-3: R2-6: The authors have shown the hydrodynamic size (Fig S6, DLS data) in PBS buffer and in culture medium. The size of the nanoparticles is expected to be larger after coating with DSPE-PEG. However, after comparing the typical sizes observed in TEM (though they are in the dried state), for nanoparticles coated with DSPE-PEG of $M_w=5000$, the size observed is substantially large in PBS (irrespective of size changing or not after storage in the medium). In the preparation, it is indicated that the nanoparticles (1 mg) coated with 10 mg DSPE-PEG by sonication for 10 minutes. The latter as they are surfactants, will also form aggregated structures and can contribute to DLS size measurements.

As there were no indications of washing steps after sonication, how did they remove the contribution from DSPE-PEG structures that were not coating the nanoparticles. The authors need to clarify this issue.

As a suggestion, if possible the authors could use cryo-TEM to see the lipid coated structure and compare the size before and after coating.

R4-3: Thanks for raising this important question. In the preparation of DSPE-PEG coated nanoparticles, we removed excessive DSPE-PEG by centrifugation and PEG-coated nanoparticles were washed by deionized water for 2-3 times. These steps have been added into the supporting information. Hence, the size of DSPE-PEG coated nanoparticles is solely contributed by surfactant coated nanoparticles rather than DSPE-PEG aggregates.

Besides, we have tried to visualize the lipid coated structure by negative staining using phosphotungstic acid. As shown in the Fig. S6a and S6b, the average size of DHBA-PPy@DSPE-PEG nanoparticles was less than 100 nm in the dry state and a small proportion of large aggregates were observed as well. For dynamic light scattering, the intensity of scattered light increased substantially when the sizes of nanoparticles increase. We inferred such variation was caused by swollen hydrophilic PEG chains and the existence of aggregates. Particularly, the small proportion of aggregates in DSPE-PEG nanoparticles can significantly raise the average hydrodynamic sizes of DHBA-PPy@DSPE-PEG nanoparticles.

Corresponding content in the revision:

Page 15, paragraph 3

“DHBA-PPy@DSPE-PEG nanoparticles were less 100 nm in their dry state (**Fig. S6a**). Also small proportion of DHBA-PPy aggregates can be observed in **Fig. S6b**. Both the hydrophilicity of PEG chains and these aggregates increase the average hydrodynamic sizes of DHBA-PPy@DSPE-PEG nanoparticles (**Fig. S6c**).”

Corresponding content in the supplementary information:

Page 2, paragraph 2

“Excessive DSPE-PEG was removed by centrifugation. DHBA-PPy@DSPE-PEG nanoparticles were subsequently washed by deionized water for 2-3 times by centrifugation.”

Fig. S6 (a) and (b)

Response R1 and R4:

Once more, many thanks to all the reviewers for all your kind questions and most helpful comments that have, without doubt, greatly improved the quality and clarity of our revised manuscript.

REVIEWER COMMENTS

Reviewer #4 (Remarks to the Author):

Please see below my comments for the different revised sections:

Regarding washing steps revision:

As washing of lipid aggregates from polymer dispersions is not a trivial step, it would be useful for the readers if the authors can provide more details of the washing steps (centrifugation duration, rcf etc.,).

Regarding revised XPS section:

- The authors claim that they followed the procedure as described in the seminal paper by Wertheim et.al. In this paper, the parameters describing the Shirley type of inelastic background are part of the fitting routine, i.e. are varied together with the line shape parameters of the core lines to minimize the difference between theoretical curve and experimental data. The authors, by explicitly not considering any background signal during fitting in fact implicitly use a constant background in their analysis (the intensity at the right side of the N1s spectra) which is rather unphysical, leaving the results again questionable.

- Some fitted components show a strong asymmetry towards the high binding energy side (see fig. S3) which is usually seen in metals (especially transition metals) but never in semiconductors (which are best described by symmetric line shapes). Consequently, the analysis presented in figure S3 is misleading, and any conclusions drawn from it must be considered highly speculative.

- In ref 30 (Zhu, M. et al.) , the authors observed charging effects. Consequently, they used the C1s core level to calibrate their energy scale ("The XPS spectra are charge corrected to the adventitious C 1s peak at 284.6 eV. "). As the materials are similar, charging effects (shifts on the energy scale) are also to be expected in the present study and, in fact, can be seen in the graph assembled by the reviewer (after digitizing the original N1s spectra presented in fig. S3). Thus, without appropriate charge correction, all results derived from fig S3 are meaningless.

In my opinion, the study is not complete and is rather at an intermediate level. Based on these aforementioned concerns, I cannot recommend this manuscript for publication.

Reviewer #4 (Remarks to the Author):

General comments (C4): In my opinion, the study is not complete and is rather at an intermediate level.

Based on these aforementioned concerns, I cannot recommend this manuscript for publication.

Response (R4): Thanks for reviewing our paper again. According to your valuable suggestions, we have made efforts to improve the analysis and conclusions about XPS data. Hope this revision can meet the qualifications for publication in *Nature Communications*.

C4-1: Regarding washing steps revision:

As washing of lipid aggregates from polymer dispersions is not a trivial step, it would be useful for the readers if the authors can provide more details of the washing steps (centrifugation duration, rcf etc.,).

R4-1: Many thanks for this suggestion. Details about washing steps have been added to the supplementary information.

Corresponding content in the supplementary information:

Page 2, paragraph 2:

“Excessive DSPE-PEG was removed by centrifugation. DHBA-PPy@DSPE-PEG nanoparticles were subsequently washed by deionized water for 2-3 times by centrifugation at 11000 rpm for 10 minutes.”

C4-2: Regarding revised XPS section:

- The authors claim that they followed the procedure as described in the seminal paper by Wertheim et.al. In this paper, the parameters describing the Shirley type of inelastic background are part of the fitting routine, i.e. are varied together with the line shape parameters of the core lines to minimize the difference between theoretical curve and experimental data.

The authors, by explicitly not considering any background signal during fitting in fact implicitly use

a constant background in their analysis (the intensity at the right side of the N1s spectra) which is rather unphysical, leaving the results again questionable.

- Some fitted components show a strong asymmetry towards the high binding energy side (see fig. S3) which is usually seen in metals (especially transition metals) but never in semiconductors (which are best described by symmetric line shapes). Consequently, the analysis presented in figure S3 is misleading, and any conclusions drawn from it must be considered highly speculative.

- In ref 30 (Zhu, M. et al.), the authors observed charging effects. Consequently, they used the C1s core level to calibrate their energy scale ("The XPS spectra are charge corrected to the adventitious C 1s peak at 284.6 eV. "). As the materials are similar, charging effects (shifts on the energy scale) are also to be expected in the present study and, in fact, can be seen in the graph assembled by the reviewer (after digitizing the original N1s spectra presented in fig. S3). Thus, without appropriate charge correction, all results derived from fig S3 are meaningless.

R4-2: Thank you for such valuable suggestions. According to your suggestions and ref 35 in this revision, we have re-processed the XPS spectra. Firstly, we calibrated the XPS spectra to the adventitious C 1s peak at 284.6 eV. Subsequently, the N 1s peaks of XPS spectra were deconvoluted into symmetric Gaussian peaks with Shirley-type background subtracting during fitting. Current researches usually deconvolute N 1s peak into 3 or 4 peaks. For both deconvolution methods, the components at ~398 eV are assigned to the quinonoid imine structure (-N=) and the components at ~400 eV are assigned to the benzenoid amine structure (-NH-). While the charged species are attributed to one peak for protonated amine at ~401 eV for "3 peaks" method or two peaks for protonated amine at ~401 eV and protonated imine at ~403 eV for "4 peaks" method. We have processed our XPS data with both deconvolution methods. For both methods, the fractions of protonated nitrogen structures exhibit a similar variation trend. According to the deconvoluted XPS spectra, imine structures contribute a large fraction of neutral nitrogen structures. Therefore, the fractions of protonated imine structures should be included during fitting and we decided to use "4 peaks" method in the revision.

Also, the deconvolution of XPS spectra can be used to explain the peak position shift of the N 1s

peaks in different PPy nanoparticles. Such phenomenon was caused by the variation of the imine structure in different PPy nanoparticles. According to previous reports, imine structure is usually caused by reduction of PPy polarons (ref 36 in this revision). In this work, phenol groups on catechol derivatives can reduce polarons and cause increase of imine structures in our catechol-PPy nanoparticles. With more polarons reduced (as shown in **Table 1**), more imine structures are formed as shown in the deconvolution of XPS spectra.

Deconvoluting N 1s peak into 3 peaks:

	-N=	-NH-	-N ⁺ = and -N ⁺ H-
Pristine	15.2%	72.1%	12.7%
DA	37.4%	50.1%	12.5%
CA	66.0%	27.2%	6.8%
DHBA	64.7%	31.4%	3.9%
DHBN	30.7%	58.9%	10.4%

Deconvoluting N 1s peak into 4 peaks (Fig S4 and Table S1):

	-N=	-NH-	-N ⁺ = and -N ⁺ H-
Pristine	20.8%	60.3%	18.9%
DA	38.3%	45.0%	16.7%
CA	69.6%	17.4%	13.0%
DHBA	78.9%	17.1%	4.0%
DHBN	41.7%	39.5%	18.8%

Corresponding content in the revision:

Page 12, paragraph 2:

“Therefore, part of polarons can be directly reduced by catechol derivatives. This process can be observed by deconvoluting the N 1s peaks in the XPS spectra of different PPy nanoparticles. The N 1s peak of PPy can be deconvoluted into four Gaussian peaks centered at ~398.4 eV for imine structure (-N=), ~399.8 eV for amine structure (-NH-), ~400.4 eV for protonated amine structure (-NH⁺-) and ~402.8 eV for protonated imine structure (-N⁺=) ³⁵. Imine structure comes from the reduction of PPy polarons ³⁶. For pristine PPy and catechol-PPy nanoparticles, the fractions of charged species (Fig. S4 and Table S1) appear a similar variation trend with the polaron content (Table 1) and the fractions of imine structure appear an opposite variation trend with the polaron content. With the relative amount of bipolarons vs. polarons increased (*R*-values in Table 1), more

polarons are reduced, yielding fractions of charged species decreased (**Table S1**). This phenomenon reveals that polaron reduction is inevitable for catechol derivatives regulating the relative amount of bipolarons vs. polarons in PPy nanoparticles. Therefore, the *R*-values can hardly vary strictly from two polarons into one bipolaron. The variation of *R*-values appears to be moderate in comparison with the polaron content (**Table 1**).”

Corresponding content in the supplementary information:

Page 3, paragraph 1:

“The XPS spectra are calibrated using C 1s peak to 284.6 eV. According to previous reports, the N 1s peaks of PPy are deconvoluted into four Gaussian peaks at 398.4 eV, 399.4 eV, 400.4 eV and 402.8 eV, which represents imine (-N=), amine (-NH-), protonated benzenoid amine (-N⁺H-), and protonated quinonoid imine (-N⁺=), respectively. The deconvolution of all XPS data was processed using XPSPEAK version 4.1 software with Shirley type background subtracting.”

Response R4:

Once more, many thanks for all your kind questions and most helpful comments that have, without doubt, greatly improved the quality and clarity of our revised manuscript.

REVIEWER COMMENTS

Reviewer #5 (Remarks to the Author):

General Comments

I was invited to provide an additional review of this manuscript; I was asked specifically to review the XPS analysis as presented, and the authors' response to concerns raised by a previous reviewer (#4). In the interests of time I will therefore only comment on XPS-specific concerns and not on the paper generally.

General comments (for more details see specific comments below)

- In my opinion, the concerns raised by reviewer #4 have been addressed satisfactorily by the authors.
- However, there are several additional concerns about the XPS analysis in this manuscript. The significance and reliability of the quantitative results (N 1s peak fitting) are questionable because of the following problems:
 - o Insufficient experimental details are provided regarding almost every aspect of the analysis, from the preparation and mounting of samples, to relevant parameters of analysis, and details of data processing, in particular peak fitting.
 - o No error analysis is presented at all.
- I'm concerned about some of the incorrect XPS terminology used by the authors. I would strongly recommend consulting the various international standards regarding XPS experimentation, analysis, and terminology. A good starting point generally would be a Special Topic Collection that was published last year by the Journal of Vacuum Science & Technology (Special Topic Collection: Reproducibility Challenges and Solutions, J. Vac. Sci. Technol. A (2020), see <https://avs.scitation.org/topic/special-collections/reprod2020?SeriesKey=jva>). It covers almost every aspect of XPS analysis with numerous references to standards and reliable literature sources.

Specific Comments (main text)

p. 6 (detecting low levels of residual Fe by XPS).

- If the authors intended to detect and quantify low concentrations of Fe they should have recorded high quality spectra, either complete survey spectra or Fe 2p narrow scans. The spectra shown in Fig. S1b are too noisy to make any meaningful statement apart from stating that Fe was not present in high concentrations at the surface. The detection limit for Fe under the analysis conditions should have been determined.

p. 12 (peak fitting the N 1s spectra).

- The authors refer to the mathematical operation of fitting XP spectra as "deconvolution". This is incorrect, the correct term is peak-fitting (or curve-fitting). Deconvolution correctly refers to a very different mathematical algorithm whereby a broadening function is removed numerically from an experimental signal such as a photoemission peak. Unfortunately, the term "deconvolution" has become widely used despite the fact it's incorrect.
- The attempt at fitting the N 1s as presented is problematic for several reasons (refer also to Supp. Info.):
 - o None of the references cited by the authors to support BE assignments are convincing, including the secondary references cited in the primary references. The peak fitting protocols, as published in those references, are of poor quality, and no paper presents any error analysis. I would instead recommend, as a starting point, a review by Kang, Teoh, and Tan from 1998 (Prog. Polym. Sci. 23 (1998) 277-324; [https://doi.org/10.1016/S0079-6700\(97\)00030-0](https://doi.org/10.1016/S0079-6700(97)00030-0)). This review is based on many years of extensive and careful work in this area, and the authors are careful in not over-interpreting XPS data. Note, however, that those authors still use a value of 284.6 eV for the aliphatic hydrocarbon C 1s peak, a value that is no longer recommended.
 - o The experimental section lacks important details. This is important because the data presented

(N 1s spectra in Fig. S4) are of poor quality (low S/N and poorly resolved). Peak fitting in a case like this can be a rather unreliable and meaningless exercise unless the fitting protocol is developed and applied with suitable care, and is comprehensively documented. For example, were any fitting constraints used and, if so, what were the constraints? The variation in peak width of some of the fitting components in Fig. S4 from one fit to another is unacceptably large, suggesting that the peak width wasn't constrained. Peak positions also appear to vary from one spectrum to another.

o The authors don't present any error analysis, either regarding XPS experimentation generally, or regarding peak fitting specifically. As a reader I have no way of assessing the reliability or reproducibility of the data and the data analysis presented. This is critically important. Almost every aspect of experimentation is a potential source of exp. uncertainty; peak fitting is particularly prone to significant uncertainties because of its rather subjective nature. So much depends on how the analyst implements and performs the fitting, and it's no surprise that peak fitting is one of the most significant sources of errors in the XPS literature (see e.g. "Assessment of the frequency and nature of erroneous x-ray photoelectron spectroscopy analyses in the scientific literature" (Journal of Vacuum Science & Technology A 38, 061204 (2020); <https://doi.org/10.1116/6.0000685>). Of particular concern for the N 1s analysis presented is the fact that the authors attempt to quantify the concentrations of four peak components that are separated by less than 1.5 eV from each other, with the spectral resolution probably being around 1.5 eV or greater. The resulting peak overlap, combined with the poor quality of the data, means that the components are highly correlated, and fit variables are poorly defined. Since the concentrations of peak components are very sensitive to peak shape, peak position and peak width, the large uncertainties in those variables would translate to even greater uncertainties in concentrations. The authors need to describe how often the analysis was repeated, either using duplicate samples or on one and the same sample and the resulting statistics (reproducibility). An error analysis should be performed for each of the N 1s fits to assess the significance and reliability of the quantitative results of the fitting process. Various methods are available and well documented in the literature, such as Monte-Carlo based methods.

o The elemental surface compositions of the samples need to be presented (atomic concentrations of C, N, O, and any other element that was detected on the surface). This is important evidence to support and confirm (or otherwise) the results obtained via fitting. Elemental compositions and peak fitting results should be cross-checked for internal consistency.

o The elemental surface compositions of the samples need to be presented (atomic concentrations of C, N, O, and any other element that was detected on the surface). This is important evidence to support and confirm (or otherwise) the results obtained via fitting. Elemental compositions and peak fitting results should be cross-checked for internal consistency.

Specific Comments (supporting information)

p. 3 (Methods section).

- Insufficient experimental detail. The peak fitting of the N 1s spectrum presented by the authors can only be justified by careful and rigorous experimentation which has to be documented. Sample preparation and mounting need to be described, data analysis conditions listed, and any other potentially relevant details mentioned, e.g., whether a charge neutralisation system has been used. How did the authors control differential charging during analysis which could potentially distort spectra? It would be very useful to see the other narrow scan spectra as well (C 1s, O 1s) because diff. charging would affect all spectra. Also, did the authors check for evidence of potential sample degradation due to X-radiation, something that is always a concern.

p. 3 (Charge correction)

- First of all, this is not a "calibration", the correct term is "charge correction". A calibration of the binding energy scale of a spectrometer is performed using known reference materials like Au, Ag, and Cu. Here the authors are describing a method of correcting a shift of the binding energy scale due to differential sample charging during analysis.

- Secondly, the value of 284.6 eV for the aliphatic hydrocarbon peak of C 1s is no longer recommended. A value of 284.8-285 eV should be used. The authors are referred to several publications that were published as part of a Special Topic Collection by the JVST (Special Topic Collection: Reproducibility Challenges and Solutions, J. Vac. Sci. Technol. A (2020), see <https://avs.scitation.org/topic/special-collections/reprod2020?SeriesKey=jva>), specifically "XPS guide: Charge neutralization and binding energy referencing for insulating samples". The authors should also consult the relevant international standards such as ASTM E1523-15, Standard Guide to Charge Control and Charge Referencing Techniques in X-Ray Photoelectron Spectroscopy (ASTM International, West Conshohocken, PA, 2015), or ISO 19318:2004 "Surface Chemical Analysis—

Reporting of Methods Used for Charge Control and Charge Correction" (International Organization for Standardization, Geneva, 2004).

- Thirdly, if the authors use the C 1s for charge referencing/correction they should show the C 1s spectrum and describe how the peak position of the reference peak was determined. This would be particularly important in this case because the C 1s spectrum of the polymer would consist of several overlapping components which would make it difficult to determine a reliable reference binding energy value.
- Fourthly, the authors need to present an error analysis regarding the measurement of binding energy values or, at the least, provide an estimate of the uncertainty associated with those measurements. In my experience, this would be at least +/- 0.2-0.3 eV. This needs to be taken into account when interpreting N 1s peak fitting results.

p. 13 (Table S1)

- What are the errors associated with these fractions? Please refer to my comment in the main text regarding error analysis.

Reviewer #5 (Remarks to the Author):

General Comments

I was invited to provide an additional review of this manuscript; I was asked specifically to review the XPS analysis as presented, and the authors' response to concerns raised by a previous reviewer (#4). In the interests of time I will therefore only comment on XPS-specific concerns and not on the paper generally.

C5: General comments (for more details see specific comments below)

- In my opinion, the concerns raised by reviewer #4 have been addressed satisfactorily by the authors.
- However, there are several additional concerns about the XPS analysis in this manuscript. The significance and reliability of the quantitative results (N 1s peak fitting) are questionable because of the following problems:
 - o Insufficient experimental details are provided regarding almost every aspect of the analysis, from the preparation and mounting of samples, to relevant parameters of analysis, and details of data processing, in particular peak fitting.
 - o No error analysis is presented at all.
- I'm concerned about some of the incorrect XPS terminology used by the authors. I would strongly recommend consulting the various international standards regarding XPS experimentation, analysis, and terminology. A good starting point generally would be a Special Topic Collection that was published last year by the Journal of Vacuum Science & Technology (Special Topic Collection: Reproducibility Challenges and Solutions, J. Vac. Sci. Technol. A (2020), see <https://avs.scitation.org/topic/special-collections/reprod2020?SeriesKey=jva>). It covers almost every aspect of XPS analysis with numerous references to standards and reliable literature sources.

R5: Firstly thanks for all the valuable suggestions about the XPS analysis. We have carefully read this Special Topic Collection about XPS analysis and their references.

Details about experiments, data processing and peak fitting have been added in the revision. Also the error analysis of peak fitting results has been presented in the supplementary information. Revisions about specific comments are provided into a point by point manner below. Hope the revised manuscript can address the concerns about XPS analysis.

Specific Comments (main text)

C5-1: p. 6 (detecting low levels of residual Fe by XPS).

- If the authors intended to detect and quantify low concentrations of Fe they should have recorded high quality spectra, either complete survey spectra or Fe 2p narrow scans. The spectra shown in Fig. S1b are too noisy to make any meaningful statement apart from stating that Fe was not present in high concentrations at the surface. The detection limit for Fe under the analysis conditions should have been determined.

R5-1: Many thanks for raising this important issue. The narrow scans of Fe 2p region for different PPy nanoparticles are added in **Fig.s1**. The characteristic Fe 2p peaks are not evident in these narrow scans as well as complete survey spectra. According to previous reports (*Ref: Surf. Interface Anal. (2014) 46, 175-185*), the detection limit of Fe in the light element matrix is about 0.1%. Thus we performed an error analysis for the atomic concentrations using Monte-Carlo simulations as shown in **Table S1**. The existence of Fe element cannot be confirmed with such large standard deviation vs. the atomic concentrations.

Since Fe elements may not present in a detectable concentration at the surface, we further used ICP-MS to determine the Fe element concentrations in the nanoparticles for reference. The mass concentrations of Fe element were about several ppm in 1 mg/mL different PPy nanoparticles (Pristine PPy: 7 ppm, DA-PPy: 4

ppm, CA-PPy: 3 ppm, DHBA-PPy: 5 ppm and DHBN-PPy: 2 ppm). Both results support the conclusion that the concentrations of Fe element in as-synthesized nanoparticles are very low.

Corresponding content in the revision:

Fig. s1c

Table S1 | Atomic concentrations calculated from the survey XPS spectra of different PPy nanoparticles. Besides C and N elements on the PPy chains, O is originated from degradation products of PPy in the ambient air. Cl is originated from the counter ions for polarons and bipolarons. And Si element is originate from the contamination in the environment like PDMS. The error analysis of atomic concentrations was performed using Monte-Carlo simulations in CasaXPS software. According to previous reports, Fe element in light element matrix is about 0.1%. Thus the existence of Fe element in PPy nanoparticles cannot be confirmed in the XPS spectra. The C/N ratio of different PPy nanoparticles varies between 7 and 12. The theoretical C/N ratio of PPy is 4 and that of DeTAB is 13. The C/N ratio of confirms the existence of residual surfactant DeTAB in the PPy nanoparticles.

	C	C-Error	N	N-Error	O	O-Error	Cl	Cl-Error	Fe	Fe-Error	Si	Si-Error	C/N
Pristine	74.75	1.06	8.70	0.93	13.35	0.6	2.77	0.3	0.04	0.08	0.39	0.56	8.6
DA	69.34	0.91	6.08	0.73	18.96	0.56	2.15	0.29	0.12	0.10	3.35	0.52	11.4
CA	70.32	0.97	6.08	0.79	18.47	0.61	1.69	0.28	0.12	0.07	3.32	0.49	11.6
DHBA	66.60	1.05	6.22	0.90	24.50	0.72	1.24	0.30	0.08	0.10	1.36	0.63	10.7
DHBN	71.25	0.99	10.01	0.80	13.35	0.53	2.20	0.34	0.08	0.09	3.11	0.55	7.1

C5-2: p. 12 (peak fitting the N 1s spectra).

- The authors refer to the mathematical operation of fitting XP spectra as “deconvolution”. This is incorrect, the correct term is peak-fitting (or curve-fitting). Deconvolution correctly refers to a very different mathematical algorithm whereby a broadening function is removed numerically from an experimental signal such as a photoemission peak. Unfortunately, the term "deconvolution" has become widely used despite the fact it's incorrect.

R5-2: Many thanks for pointing out this error. We have replaced “deconvolution” by “peak fitting” in the revision.

Corresponding content in the revision:

Page 12, paragraph 2:

“The fraction of imine quinonoid structure can be estimated by fitting the N 1s peaks (details are shown in the supplementary information) in the XPS spectra pristine and catechol-PPy nanoparticles.”

C5-3: The attempt at fitting the N 1s as presented is problematic for several reasons (refer also to Supp. Info.):

C5-3-1: None of the references cited by the authors to support BE assignments are convincing, including the secondary references cited in the primary references. The

peak fitting protocols, as published in those references, are of poor quality, and no paper presents any error analysis. I would instead recommend, as a starting point, a review by Kang, Teoh, and Tan from 1998 (*Prog. Polym. Sci.* 23 (1998) 277-324; [https://doi.org/10.1016/S0079-6700\(97\)00030-0](https://doi.org/10.1016/S0079-6700(97)00030-0)). This review is based on many years of extensive and careful work in this area, and the authors are careful in not over-interpreting XPS data. Note, however, that those authors still use a value of 284.6 eV for the aliphatic hydrocarbon C 1s peak, a value that is no longer recommended.

R5-3-1: Thanks for recommending this important review. According to this review and its references, we used similar binding energy (BE) assignments for fitting N 1s peak in our XPS analysis. In this review, the BE assignments were performed based on the neutral C 1s peak at 284.6 eV. For pure aromatic polymers like polyaniline and polypyrrole, the BE of aromatic hydrocarbon is more appropriately assigned to 284.7 eV (*Ref: J. Vac. Sci. Technol. A (2020) 38, 023207, ISO 19318:2004, Surf. Interface Anal. (2010) 42, 1184-1187*). Therefore, the hydrocarbon on PPy molecules can be used as the internal reference for charge correction. Accordingly, the BEs of quinonoid imine and benzenoid are assigned to 398.5 eV and 399.7 eV, respectively. Also the high BE tail over 400 eV has a multiple contribution of species, including protonated imine/amine, surface oxidation products, interchain H-bonding, and the imine nitrogen satellite (*Ref: J. Vac. Sci. Technol. A 38, 031002 (2020), Prog. Polym. Sci. (1998) 23, 277-324*). Thus we only use two peaks separated by about 1.5 and 3.0 eV from amine peak for purposes of tracking the peak shape changes. No attempt to assign specific species to each peak will be made.

For our PPy nanoparticles, the neutral C 1s peak is more appropriately corrected to 284.9 eV accounting for both aromatic (284.7 eV) and aliphatic hydrocarbon (285.0 eV) together. The aromatic hydrocarbon originates from PPy backbone and the aliphatic hydrocarbon originates from residual surfactant DeTAB (Decyltrimethylammonium bromide). The existence of residual DeTAB can be

confirmed by the C/N atom ratio from the XPS spectra of our PPy nanoparticles. The theoretical C/N ratio for polypyrrole is 4 and that ratio for DeTAB is 13. The C/N ratio of pristine and catechols-PPys nanoparticles measured by XPS varies from 7 to 12 (as shown in **Table S1**). Since aromatic and aliphatic carbon give rise two peaks being very close, one single component peak is recommended to be used for accounting both functionalities together. Thus, we chose 284.9 eV for charge correction in analyzing our PPy nanoparticles.

C5-3-2:

o The experimental section lacks important details. This is important because the data presented (N 1s spectra in Fig. S4) are of poor quality (low S/N and poorly resolved). Peak fitting in a case like this can be a rather unreliable and meaningless exercise unless the fitting protocol is developed and applied with suitable care, and is comprehensively documented. For example, were any fitting constraints used and, if so, what were the constraints? The variation in peak width of some of the fitting components in Fig. S4 from one fit to another is unacceptably large, suggesting that the peak width wasn't constrained. Peak positions also appear to vary from one spectrum to another.

R5-3-2: Many thanks for pointing this critical issue. We have re-processed the peak fitting of N 1s peak using 284.9 eV as references. The detailed fitting protocol has been added into the supporting information. The variation of position, line shape and linewidth of fitting components was kept similar for all fitted curves.

Corresponding content in the supplementary information:

Page 3, paragraph 3:

Data processing was performed using CASAXPS processing software version 2.3.24 (Casa Software Ltd., Teignmouth, UK). All elements present were identified from survey spectra. The atomic concentrations of the detected elements were calculated using integral peak intensities and the sensitivity factors supplied by the

manufacturer. Binding energies were referenced to the C 1s peak at 284.9 eV (combination of aromatic and aliphatic hydrocarbon) with an uncertainty of ± 0.2 eV. For fitting the N 1s peak, we use a slightly modified Universal Polymer Tougaard background and pseudo-Voigt functions (LA line shapes in CasaXPS) for the component peaks. Slightly asymmetric should be accounted for aromatic structures. This is because a proportion of the photon energy may contribute to atomic oscillations associated with the bonds within a molecule and is not available to the photoelectron as kinetic energy. The consequence of these alternative energy transfer mechanisms is that the XPS peaks exhibit asymmetry (*Ref: J. Vac. Sci. Technol. A (2021) 39, 013204, CasaXPS Manual 2.3.24: Asymmetry of Peaks in the XPS of Polymers*). The full width at half maximum (FWHM) is constrained to a narrow range of 1.4–1.8 eV. We use four components at 398.5 ± 0.2 eV (quinonoid imine, -N=), 399.7 ± 0.2 eV (benzenoid amine, -NH-), and two peaks at about 401.2 ± 0.2 eV and 402.7 ± 0.2 eV for purposes of tracking the peak shape changes. The high BE tail above 400 eV can be contributed by various species including protonated imine/amine, surface oxidation products, interchain H-bonding, and the imine nitrogen satellite. Thus no attempt to assign specific species to each peak will be made (*Ref: J. Vac. Sci. Technol. A 38, 031002 (2020), Prog. Polym. Sci. (1998) 23, 277-324.*). The error analysis of peak fitting results is performed using Monte-Carlo simulations in CASAXPS. The standard deviation of the peak area based on peak fitting results are about 1%-5%.

C5-3-3:

o The authors don't present any error analysis, either regarding XPS experimentation generally, or regarding peak fitting specifically. As a reader I have no way of assessing the reliability or reproducibility of the data and the data analysis presented. This is critically important. Almost every aspect of experimentation is a potential source of exp. uncertainty; peak fitting is particularly prone to significant uncertainties because of its rather subjective nature. So much depends on how the

analyst implements and performs the fitting, and it's no surprise that peak fitting is one of the most significant sources of errors in the XPS literature (see e.g. "Assessment of the frequency and nature of erroneous x-ray photoelectron spectroscopy analyses in the scientific literature" (Journal of Vacuum Science & Technology A 38, 061204 (2020); <https://doi.org/10.1116/6.0000685>). Of particular concern for the N 1s analysis presented is the fact that the authors attempt to quantify the concentrations of four peak components that are separated by less than 1.5 eV from each other, with the spectral resolution probably being around 1.5 eV or greater. The resulting peak overlap, combined with the poor quality of the data, means that the components are highly correlated, and fit variables are poorly defined. Since the concentrations of peak components are very sensitive to peak shape, peak position and peak width, the large uncertainties in those variables would translate to even greater uncertainties in concentrations. The authors need to describe how often the analysis was repeated, either using duplicate samples or on one and the same sample and the resulting statistics (reproducibility). An error analysis should be performed for each of the N 1s fits to assess the significance and reliability of the quantitative results of the fitting process. Various methods are available and well documented in the literature, such as Monte-Carlo based methods.

R5-3-3: Many thanks for such helpful suggestions. The details about peak fitting are added in **R5-3-2**. In the revision, we chose to use Monte-Carlo based tests to assess the significance of the peak fitting results. The overlays of all the simulated experimental spectra generated by a Monte Carlo analysis have been added as follows. The relatively high signal-to-noise ratios of measured N 1s peaks cause the variation of peak area around 1%-5%. The standard deviations for other parameters have also been added in the following table. The fraction of quinonoid imine exhibited significant increasing in DA/CA/DHBA- PPy nanoparticles compared with that in pristine and DHBA PPy nanoparticles.

Peak fitting results of the N 1s region and simulated spectra generated by Monte-Carlo simulations

Pristine

DA

CA

DHBA

DHBN

Error analysis based on the Monte-Carlo matrix

		Pristine	DA	CA	DHBA	DHBN
Amine	Position (eV)	0.024	0.025	0.076	0.068	0.031
	FWHM (Ev)	0.037	0.039	0.114	0.104	0.040
Charged and shake up-1	Position (eV)	0.048	0.060	0.086	0.078	0.081
	FWHM (Ev)	0.055	0.111	0.152	0.154	0.075
Charged and shake up-2	Position (eV)	0.068	0.079	0.166	0.174	0.085
	FWHM (Ev)	0.153	0.091	0.186	0.172	0.153
Imine	Position (eV)	0.014	0.045	0.023	0.026	0.022
	FWHM (Ev)	0.059	0.074	0.082	0.054	0.077

Corresponding content in the revision:

Page 12, paragraph 2

“Therefore, part of polarons can be directly reduced by catechol derivatives. Reduced polarons exhibit imine quinonoid structure (-N=). The variation of imine quinonoid structure can be estimated by analyzing the XPS peaks of PPy nanoparticles. The fractions of imine quinonoid structure can be estimated by fitting the N 1s peaks (details are shown in the supplementary information) in the XPS spectra pristine and catechol-PPy nanoparticles. For DA/CA/DHBA- PPy nanoparticles with stronger T₂ MRI contrasts (T₂ relaxation times in **Table 1**), significant increase of the reduced polarons (imine quinonoid structure) can also be observed in these nanoparticles as well as decrease of polaron content (**Fig. S4**). This phenomenon reveals that polaron reduction is inevitable for catechol derivatives regulating the relative amount of bipolarons vs. polarons in PPy nanoparticles. Also, the fractions of reduced polarons exhibit little variation between pristine and DHBN- PPy nanoparticles (**Fig. S4**). This result further supports above speculations that DHBN has very weak ability for trapping polaron radicals.”

Fig.s4| Peak fitting analysis of XPS N 1s peak of (a) pristine, (b) DA, (c) CA, (d) DHBA and (e) DHBN PPy nanoparticles. Fractions of amine (f) and imine (g) structures estimated by peak fitting analysis. The standard deviation of peak area is estimated by Monte-Carlo simulations in CasaXPS software.

R5-3-4:

o The elemental surface compositions of the samples need to be presented (atomic concentrations of C, N, O, and any other element that was detected on the surface). This is important evidence to support and confirm (or otherwise) the results

obtained via fitting. Elemental compositions and peak fitting results should be cross-checked for internal consistency.

C5-3-4: Thanks for raising this important question. Detected elements include C, N, O, Cl and Si. C and N atoms come from the PPy chain and Cl atoms come from the counter ions for protonated N atoms on polarons and bipolarons. O atoms come from the degradation of PPy by oxygen in the ambient air and Si come from the contamination in the environment. As explained above, the C/N atomic ratios of our PPy nanoparticles (7-12) indicate the existence of DeTAB residuals. The atomic concentration of Cl can be used to estimate the total protonation level of PPy nanoparticles. The reduction of Cl concentrations can also be attributed to the reduction of polarons by catechol derivatives as that of imine fraction. The similar variation of Cl concentrations and imine fractions supports that internal consistency of the peak fitting analysis.

Specific Comments (supporting information)

C5-4: p. 3 (Methods section).

- Insufficient experimental detail. The peak fitting of the N 1s spectrum presented by the authors can only be justified by careful and rigorous experimentation which has to be documented. Sample preparation and mounting need to be described, data analysis conditions listed, and any other potentially relevant details mentioned, e.g., whether a charge neutralisation system has been used. How did the authors control differential charging during analysis which could potentially distort spectra? It would be very useful to see the other narrow scan spectra as well (C 1s, O 1s) because differential charging would affect all spectra. Also, did the authors check for evidence of potential sample degradation due to X-radiation, something that is always a concern.

R5-4: Thanks for pointing this important issue. Firstly, the experimental details about sample preparation, mounting and data analysis conditions have been added into the supplementary information. Then, the charging effects for PPy are expected to be minimal because PPy is a conductive polymer (*Ref: J. Vac. Sci. Technol. A 39, 013204 (2021), Surf. Inter. Anal. 11, 327 (1988)*). Positively charged sample surface can be quickly neutralized by the flow of delocalized or free electrons from the bulk to the surface. To minimize the charging effect, we grounded the samples by making electrical contact between the sample surface and the sample mount by silicon-free double-sided carbon (*Ref: J. Vac. Sci. Technol. A 38, 031204 (2020)*). Also, please find the narrow scan spectra of C 1s and O 1s region below.

Besides the influence of sample degradation is also checked by repeating measurements of the same sample. Usually sample degradation caused by X-ray irradiation is slow after exposure times less than 1 hour (*Ref: High Resolution XPS of Organic Polymers: The Scienta ESCA300 Database (Wiley, Chichester, 1992)*). Here we increased the sweep times of 10 minutes to 30 minutes for testing the potential degradation effect. For all PPy nanoparticles, the peak shapes (regardless the intensities and the positions) of N 1s region remain identical between two repeated measurements. These results suggest the potential sample degradation caused by x-ray irradiation have negligible effects on the peak fitting analysis of the N 1s region. While for the peak shapes of C 1s region, only CA- and DHBA- PPy nanoparticles exhibit slight variation on the line width, which might originates from sample degradation.

Narrow scans of the C 1s and N 1s region of the same sample.

Repeated narrow scans of C 1s and N 1s region of the same sample. The red lines represent the first sweep and the green lines represent the second sweep.

Corresponding content in the supplementary information:

Page 3, paragraph 2

"X-ray photoelectron spectroscopy (XPS) analysis was performed using a PHI 5000C ESCA System (Perkin-Elmer Inc., US) with a Mg K α achromatic X-ray source (1253.6 eV) at a power of 250 W, 14 kV. The total pressure in the main vacuum chamber during analysis was typically between 10^{-7} and 10^{-8} Pa. Survey spectra were acquired at a pass energy of 93.9 eV and a step size of 1 eV. To obtain more detailed information about chemical structure, oxidation states etc., high-resolution spectra were recorded from individual peaks at 93.9 eV pass energy and 0.2 eV step size. Each specimen was analyzed at an emission angle of 0° as measured from the surface normal. The synthesized nanoparticles were washed with ethanol and dried under vacuum. Samples for XPS measurement were prepared by compressing about 15 mg of dried nanoparticles into conductive carbon tapes. Also the sample surfaces were also grounded with the sample mount by conductive carbon tapes for minimizing potential charging effect.

Data processing was performed using CASAXPS processing software version 2.3.24 (Casa Software Ltd., Teignmouth, UK). All elements present were identified from survey spectra. The atomic concentrations of the detected elements were calculated using integral peak intensities and the sensitivity factors supplied by the manufacturer. Binding energies were referenced to the C 1s peak at 284.9 eV (combination of aromatic and aliphatic hydrocarbon at 284.7 eV and 285.0 eV, respectively) with an uncertainty of ± 0.2 eV. For fitting the N 1s peak, we used a slightly modified Universal Polymer Tougaard background and pseudo-Voigt functions (LA line shapes in CasaXPS) for the component peaks. Slightly asymmetric should be accounted for aromatic structures. This is because a proportion of the photon energy may contribute to atomic oscillations associated with the bonds within a molecule and is not available to the photoelectron as kinetic energy. The consequence of these alternative energy transfer mechanisms is that the XPS peaks

exhibit asymmetry. The full width at half maximum (FWHM) is constrained to a narrow range of 1.4–1.8 eV. We use four components at 398.5±0.2 eV (quinonoid imine, -N=), 399.7±0.2 eV (benzenoid amine, -NH-), and two peaks at about 401.2±0.2 eV and 402.7±0.2 eV for purposes of tracking the peak shape changes. The high BE tail above 400 eV can be contributed by various species including protonated imine/amine, surface oxidation products, interchain H-bonding, and the imine nitrogen satellite. Thus no attempt to assign specific species to each peak will be made. The error analysis of peak fitting results is performed using Monte-Carlo simulations in CASAXPS. The standard deviation of the peak area based on peak fitting results are about 1%-5%.”

C5-5-1: p. 3 (Charge correction)

- First of all, this is not a "calibration", the correct term is "charge correction". A calibration of the binding energy scale of a spectrometer is performed using known reference materials like Au, Ag, and Cu. Here the authors are describing a method of correcting a shift of the binding energy scale due to differential sample charging during analysis.

R5-5: Many thanks for pointing this error. We have replaced “calibration” with “charge correction” in the supplementary information.

C5-5-2&5-5-3:

- Secondly, the value of 284.6 eV for the aliphatic hydrocarbon peak of C 1s is no longer recommended. A value of 284.8-285 eV should be used. The authors are referred to several publications that were published as part of a Special Topic Collection by the JVST (Special Topic Collection: Reproducibility Challenges and Solutions, J. Vac. Sci. Technol. A (2020), see <https://avs.scitation.org/topic/special-collections/reprod2020?SeriesKey=jva>), specifically "XPS guide: Charge neutralization and binding energy referencing for insulating samples". The authors should also

consult the relevant international standards such as ASTM E1523-15, Standard Guide to Charge Control and Charge Referencing Techniques in X-Ray Photoelectron Spectroscopy (ASTM International, West Conshohocken, PA, 2015), or ISO 19318:2004 “Surface Chemical Analysis—Reporting of Methods Used for Charge Control and Charge Correction” (International Organization for Standardization, Geneva, 2004).

- Thirdly, if the authors use the C 1s for charge referencing/correction they should show the C 1s spectrum and describe how the peak position of the reference peak was determined. This would be particularly important in this case because the C 1s spectrum of the polymer would consist of several overlapping components which would make it difficult to determine a reliable reference binding energy value.

R5-5-2&5-5-3: Many thanks for pointing this important issue. In the revision we use 284.9 eV as the combination of aromatic and aliphatic hydrocarbon by consulting above references (please refer to previous responses). The C/N ratio of our PPy nanoparticles ranges between the theoretical C/N ratio of PPy (4) and surfactant DeTAB (13). The BE of aromatic hydrocarbon on PPy should be corrected to 284.7 eV and that of aliphatic hydrocarbon on DeTAB should be corrected to 285.0 eV. In consideration of the overlap of these two peaks, we decided to use one single component accounting for both aromatic and aliphatic hydrocarbon according to previous researches (*Refs: J. Vac. Sci. Technol. A 38, 023207 (2020), J. Vac. Sci. Technol. A 39, 013204 (2021)*).

R5-5-4: • Fourthly, the authors need to present an error analysis regarding the measurement of binding energy values or, at the least, provide an estimate of the uncertainty associated with those measurements. In my experience, this would be at least +/- 0.2-0.3 eV. This needs to be taken into account when interpreting N 1s peak fitting results.

R5-5-4: Thanks for raising this important question. To estimate the uncertainty of

measurements, we used rapid and repeated narrow scans for the C 1s region. As explained above, the charging effect and the degradation only produce negligible effect on the peak shape of C 1s region for pristine/DA/DHBN PPy nanoparticles. Thus we can use the variation of peak position for C 1s peak to estimate the uncertainty associated with those measurements. The variation of the peak position for these samples ranges about 0.4 eV. Accordingly, we chose ± 0.2 eV as the uncertainty of the measurements.

C5-6: p. 13 (Table S1 in the previous revision)

- What are the errors associated with these fractions? Please refer to my comment in the main text regarding error analysis.

R5-6: Thanks again for pointing the issues about error analysis. Error analysis (please refer to previous responses.) about atomic concentrations and peak fitting parameters (position, area and linewidth) based on Monte-Carlo method has been added into the supplementary information.

Response R5:

Once more, many thanks for all your kind questions and most helpful comments that have, without doubt, greatly improved the quality and clarity of our revised manuscript.

REVIEWERS' COMMENTS

Reviewer #5 (Remarks to the Author):

I am quite overwhelmed by the amount of effort and care the authors have taken to address the concerns I had raised in my original review. This is easily the best and most satisfactory response by any author to one of my reviews, and I have reviewed many manuscripts in my 35 year career. I sincerely congratulate the authors and thank them for taking my advice so seriously. There is obviously a willingness on their part to learn and improve their competency as scientist; this is a very welcome attitude and is fundamental how we can advance the quality and integrity as science in general.

I have added just a few comments to the SI where I think it would be useful to include a couple of clarifying remarks (annotated document attached) but otherwise this manuscript can be published as is.

Again, I'd like to thank the authors for having taken a lot of time to respond to my concerns, I think (at least, I hope) the manuscript is now in a much better shape.

Good luck in your future work!

Reviewer #5:

General Comments: I am quite overwhelmed by the amount of effort and care the authors have taken to address the concerns I had raised in my original review. This is easily the best and most satisfactory response by any author to one of my reviews, and I have reviewed many manuscripts in my 35 year career. I sincerely congratulate the authors and thank them for taking my advice so seriously. There is obviously a willingness on their part to learn and improve their competency as scientist; this is a very welcome attitude and is fundamental how we can advance the quality and integrity as science in general.

I have added just a few comments to the SI where I think it would be useful to include a couple of clarifying remarks (annotated document attached) but otherwise this manuscript can be published as is.

Again, I'd like to thank the authors for having taken a lot of time to respond to my concerns, I think (at least, I hope) the manuscript is now in a much better shape.

Good luck in your future work!

Response: Many thanks for your positive comments. The description about XPS analysis are moved into the Method and clarifying remarks in blue-highlighted are added.

Corresponding content in the Method section:

“Each specimen was analyzed at an emission angle of 0° as measured from the surface normal.” was removed because microscopic angle for powder analysis would vary between 0 and 90 degrees

“high-resolution spectra” was replaced by “narrow scans” with the same pass energy used.

“It should be noted that the standard deviations are correlated to the constraints of peak fitting parameters. Tight constraints can reduce the standard deviations.” are added into the method section.

Response R5:

Once more, many thanks for all your kind questions and most helpful comments that have, without doubt, greatly improved the quality and clarity of our revised manuscript.